# Global sensitivities of reactive N and S gas and particle concentrations and deposition to precursor emissions reductions

Yao Ge[1,2,a], Massimo Vieno[2], David S. Stevenson[3], Peter Wind[4], Mathew R. Heal[1]

[1] School of Chemistry, University of Edinburgh, Joseph Black Building, David Brewster Road, Edinburgh, EH9 3FJ, UK

[2] UK Centre for Ecology & Hydrology, Bush Estate, Penicuik, Midlothian, EH26 0QB, UK

[3] School of GeoSciences, University of Edinburgh, Crew Building, Alexander Crum Brown Road, Edinburgh, EH9 3FF, UK

[4] The Norwegian Meteorological Institute, Henrik Mohns Plass 1, 0313, Oslo, Norway

[a] Now at: The Norwegian Meteorological Institute, Henrik Mohns Plass 1, 0313, Oslo, Norway

*Correspondence to*: Yao Ge (Y.Ge-7@sms.ed.ac.uk), Mathew R. Heal (M.Heal@ed.ac.uk)

**Abstract.** The reduction of fine particles ($PM_{2.5}$) and reactive N ($N_r$) and S ($S_r$) species is a key objective for air pollution control policies because of their major adverse effects on human health, ecosystem diversity, and climate. The sensitivity of global and regional $N_r$, $S_r$, and $PM_{2.5}$ to 20% and 40% individual and collective reductions in anthropogenic emissions of $NH_3$, $NO_x$, and $SO_x$ (with respect to a 2015 baseline) is investigated using the EMEP MSC-W atmospheric chemistry transport model with WRF meteorology. Regional comparisons reveal that the individual emissions reduction has multiple co-benefits and small disbenefits on different species, and those effects are highly geographically variable. A 40% $NH_3$ emission reduction decreases regional average $NH_3$ concentrations by 47-49%, but only decreases $NH_4^+$ by 18% in Euro_Medi, 15% in East Asia, 12% in North America, and 4% in South Asia. This order follows the regional ammonia-richness. A disbenefit is the increased $SO_2$ concentrations in these regions (10-16% for 40% reductions) because reduced $NH_3$ levels decrease $SO_2$ deposition through altering atmospheric acidity. A 40% $NO_x$ emission reduction reduces $NO_x$ concentrations in East Asia by 45%, Euro_Medi and North America by ~38%, and South Asia by 22%, whilst the regional order is reversed for fine $NO_3^-$, which is related to enhanced $O_3$ levels in East Asia (and also, but by less, in Euro_Medi), and decreased $O_3$ levels in South Asia (and also, but by less, in North America). Consequently, the oxidation of $NO_x$ to $NO_3^-$ and of $SO_2$ to $SO_4^{2-}$ is enhanced in East Asia but decreased in South Asia, which causes a less effective decrease in $NO_3^-$ and even an increase in $SO_4^{2-}$ in East Asia, but quite the opposite in South Asia. For regional policy making, it is thus vital to reduce three precursors together to minimise such adverse effects. A 40% $SO_x$ emission reduction is slightly more effective in reducing $SO_2$ (42-45%) than $SO_4^{2-}$ (34-38%), whilst the disbenefit is that it yields ~12% increase in $NH_3$ total deposition in the four regions which further threatens ecosystem diversity. This work also highlights important messages for policy-makers concerning the mitigation of $PM_{2.5}$. More emissions controls focusing on $NH_3$ and $NO_x$ are necessary for regions with better air quality such as northern Europe and eastern North America. In East Asia, the three individual reductions are equally effective, whilst in South Asia only $SO_x$ reduction is currently effective. The geographically-varying non-one-to-one proportionality of chemical responses of $N_r$, $S_r$, and $PM_{2.5}$ to emissions reductions revealed by this work show the importance of both prioritising emissions strategies in different regions and combining several precursor reductions together to maximise the policy effectiveness.

# 1 Introduction

Reactive N ($N_r$) and S ($S_r$) species are critical determinants of air quality. A substantial proportion of ambient $PM_{2.5}$ (particulate matter with aerodynamic diameter ≤2.5 μm) is secondary inorganic aerosol (SIA) formed from chemical reactions of emissions of the precursor gases $NH_3$, $NO_x$ (NO and $NO_2$) and $SO_x$ (sulfur oxides, mainly $SO_2$) (Behera et al., 2013; Weber et al., 2016; Vasilakos et al., 2018; Nenes et al., 2020). $PM_{2.5}$ is consistently associated with elevated risk of all-cause mortality and other adverse health impacts (Hart et al., 2015; Chen et al., 2017; Chen et al., 2018a; Karimi et al., 2019; Stieb et al., 2020). The gases $NO_2$ and $SO_2$ are also direct health pollutants. In addition, oxidized N (e.g., $NO_x$, $HNO_3$, and $NO_3^-$, collectively abbreviated as OXN) and reduced N (e.g., $NH_3$ and $NH_4^+$, collectively abbreviated as RDN) species are powerful nutrients for plants and microorganisms, whose deposition leads to eutrophication and loss of ecosystem biodiversity (Erisman et al., 2005; Bergström and Jansson, 2006; Sun et al., 2017; Kharol et al., 2018). The severity of the adverse effects of N deposition is determined not only by the total quantity but also by its form. Many studies show different N deposition components have varying toxicity to different plants, dry deposition of $NH_3$ is particularly deleterious for example (Van Herk et al., 2003; Sheppard et al., 2011; Sutton et al., 2014, 2020; Pescott et al., 2015) Deposition of oxidized S (i.e., $SO_2$ and $SO_4^{2-}$, collectively abbreviated as OXS) also greatly influences precipitation acidity (Lu et al., 2010; Aas et al., 2019; McHale et al., 2021). These observations put greater emphasis on mitigation of certain deposition components.

The East Asia, South Asia, Euro_Medi (Europe and Mediterranean) and North America regions have high population density and high $N_r$ and $S_r$ pollution. Historically, Europe and North America were the dominant emissions regions, suffering severe air pollution until the late 20th century. As reductions in $SO_x$ and $NO_x$ emissions took effect in Europe and North America, emissions in East and South Asia increased dramatically due to rapid industrialisation and dominated global $N_r$ and $S_r$ emissions by the early 21st century (Weber et al., 2016; NEC, 2019; Fowler et al., 2020), although China, in particular, is now implementing effective $SO_x$ and $NO_x$ emissions controls (Liu et al., 2016; Hoesly et al., 2018; Zheng et al., 2018; Meng et al., 2022). In contrast, a lack of action on $NH_3$ emissions in most countries, coupled with the growth in agriculture to feed a rising global population, means that global $NH_3$ emissions continue to grow (Heald et al., 2012; Fowler et al., 2015; Aksoyoglu et al., 2020). As a result, ambient $N_r$ and $S_r$ pollution remains a major health and environmental concern in most regions. The European Environmental Agency reported that 97% of the urban population in the European Union in 2019 was exposed to annual mean concentrations of $PM_{2.5}$ above the latest World Health Organization (WHO) air quality guideline of 5 μg m$^{-3}$ (EEA, 2021; WHO, 2021), whilst the United States Environmental Protection Agency reported that for the period 2014-2016 only 10% of its 429 monitoring sites had $PM_{2.5}$ concentrations <6.0 μg m$^{-3}$ (USEPA, 2017). Combining satellite retrievals, chemistry model simulations, and ground level measurements, Ma et al. (2014) and Brauer et al. (2016) showed that in 2013 the majority of the East and South Asia population lived in areas where annual mean $PM_{2.5}$ concentrations exceeded the WHO Interim Target 1 of 35 μg m$^{-3}$ (WHO, 2021). Furthermore, as mitigation of $NO_x$ emissions in recent years has been more effective than for $NH_3$, deposition of RDN is now increasingly responsible for the exceedances of N critical loads for eutrophication in many regions (Jovan et al., 2012; Chen et al., 2018b; Simpson et al., 2020; Yi et al., 2021; Jonson et al., 2022).

Understanding the sensitivities of $PM_{2.5}$, $N_r$ and $S_r$ pollution to emissions reductions is complicated not only by the substantial regional heterogeneity in relative emissions but also by the substantial meteorological heterogeneity influencing the chemistry and deposition. This necessitates use of atmospheric chemistry transport models (ACTMs) designed to simulate the underlying physical–chemical processes linking emissions, dispersion, chemical reactions, and deposition of atmospheric components. Previous ACTM studies have provided insight into the complexities of sensitivities of $PM_{2.5}$ and its SIA components to changes in emissions in different regions that measurements cannot reveal. For example, using the GEOS-Chem model, Wang et al. (2013) showed that SIA concentrations in 2015 decreased in South China and Sichuan Basin but increased in North China compared to their 2006 levels in response to −16% $SO_2$ and +16% $NO_x$ emissions changes (no change in $NH_3$ emissions) from 2006 to 2015 according to China's 12th Five-Year Plan, but if $NH_3$ emissions increase by +16% (based

on their growth rate from 2006 to 2015), the SIA reduction due to $SO_2$ reduction will be totally offset in all regions because of the elevated $NH_3NO_3$ formation, demonstrating the importance of $NH_3$ control on China's SIA mitigation. Pommier et al. (2018) reported substantial projected growth in emissions in India between 2011 and 2050, amounting to 304% for $SO_x$, 287% for NMVOC, 162% for $NO_x$, 100% for primary $PM_{2.5}$, and 60% for CO and $NH_3$, leading to increases in annual mean $PM_{2.5}$ and $O_3$ concentrations of 67% and 13% respectively. In the UK, results from EMEP4UK model simulations for 2010 emissions and meteorology indicated that $NH_3$ emissions reductions are the most effective single-component control (compared to individual reductions in $NO_x$, $SO_x$, and primary $PM_{2.5}$) on area-weighted $PM_{2.5}$, whilst weighting by population placed greater emphasis on reductions in emissions of primary $PM_{2.5}$ (Vieno et al., 2016). Holt et al. (2015) used GEOS-Chem to investigate $PM_{2.5}$ sensitivities in the United States to emissions reductions between two sets of scenarios representing a 2005 baseline (high emissions) and a 2012 analogue (low emissions). They found larger sensitivities of $PM_{2.5}$ to $SO_x$ and $NO_x$ controls in the low emissions case since lower $NO_x$ emissions in 2012 enhance the relative importance of aqueous-phase $SO_2$ oxidation.

These studies analysed $N_r$ and $S_r$ responses to precursor emissions reductions in the early 2000s and in specific regions, but do not provide a global view of sensitivities to the same reductions everywhere. Given the considerable emissions changes in global and regional $NH_3$, $NO_x$, and $SO_x$ in recent years (Hoesly et al., 2018; Kurokawa and Ohara, 2020), our understanding of the current chemical climate for $N_r$ and $S_r$ reactions on the global and regional scale and how it affects responses of $PM_{2.5}$, $N_r$ and $S_r$ species to various emissions reductions should be updated. This is the motivation for the work presented here, which provides a global picture of the effectiveness of $NH_3$, $NO_x$, and $SO_x$ emissions reductions for mitigating both concentrations and deposition of $N_r$ and $S_r$ pollutants. We used the EMEP MSC-W ACTM to simulate the global domain based on global emissions and meteorology in 2015, which enables a regional comparison to be conducted with inherently consistent simulations. The focus here is on annual means, as these are the long-term metric within global and regional air quality standards. We first describe the model set-up and performance and the sensitivity experiments used to simulate responses of $PM_{2.5}$, $N_r$ and $S_r$ species to 20% and 40% reductions in gaseous precursor emissions (Sect. 2). Section 3 details the global and regional concentration and deposition changes in components of RDN, OXN, OXS, and $PM_{2.5}$ between baseline and emissions reduction scenarios. Section 4 discusses key processes that determine the benefits and disbenefits of emissions reductions and how they vary geographically, and the implications of our findings for policy making.

## 2 Methods

### 2.1 Model set-up and performance

The EMEP MSC-W (European Monitoring and Evaluation Programme Meteorological Synthesizing Centre – West) open-source atmospheric chemistry transport model (https://www.emep.int, last access: 8 August 2022) is a three-dimensional Eulerian model widely used for both scientific research and policy development (Bergström et al., 2014; Jonson et al., 2017; Pommier et al., 2018; McFiggans et al., 2019; Karl et al., 2019; Pommier et al., 2020; Jonson et al., 2022). Version rv4.34 was used here. A detailed technical description of EMEP MSC-W rv4.0 is documented in Simpson et al. (2012). A series of overviews of model updates from version rv4.0 to rv4.34 is documented in annual EMEP status reports (Simpson et al., 2013; Tsyro et al., 2014; Simpson et al., 2015, 2016, 2017, 2018, 2019, 2020b). Meteorology for 2015 was derived from the Weather Research and Forecasting model (WRF; https://www.wrf-model.org; https://github.com/wrf-model/WRF/releases/tag/v4.2.2, last access: 8 August 2022) version 4.2.2. The coupled EMEP-WRF system has been tested and applied to many regional and global studies (Vieno et al., 2010, 2014, 2016; Werner et al., 2018; Chang et al., 2020; Gu et al., 2021).

Detailed global EMEP-WRF configurations used in this work are presented in Ge et al. (2021b, 2022). In brief, the global domain has a horizontal resolution of $1° \times 1°$ and 21 terrain-following vertical layers from the surface up to 100 hPa. The height of the lowest model layer is around 45 m. The model outputs of surface concentrations are adjusted to correspond to 3 m above ground level in order to provide concentrations at heights more typical of ambient measurements and human exposure

(Simpson et al., 2012). The aerosol module is the Equilibrium Simplified Aerosol Model V4 (EQSAM4clim), which parameterizes a full gas–liquid–solid partitioning scheme for semi-volatile and non-volatile mixtures. Details are described in Metzger et al. (2016, 2018). Dry deposition of gaseous species and aerosol components to the ground surface is simulated utilizing deposition velocity as described in Simpson et al. (2012, 2020). The parameterisation of wet deposition incorporates both in-cloud and below-cloud scavenging of gases and particles (Berge and Jakobsen, 1998; Simpson et al., 2012).

The global model evaluation of $N_r$ and $S_r$ concentrations and wet deposition from this model configuration for 2010 and 2015 against measurements from 10 ambient monitoring networks is documented in Ge et al. (2021b) and demonstrates the model's capability for capturing the spatial and seasonal variations of $NH_3$, $NH_4^+$, $NO_2$, $HNO_3$, $NO_3^-$, $SO_2$, and $SO_4^{2-}$ in East Asia, Southeast Asia, Europe, and North America. For instance, the correlation coefficients between global model and measurement annual mean concentrations for most species in 2015 are $\geq 0.78$, and for annual wet deposition of RDN and OXN are 0.78 and 0.63 respectively. This is in spite of inherent uncertainty in both model and measurements and differences in their spatial representativeness. Section S1 in the Supplement also gives a brief introduction of the model performance compared to measurements for RDN species, as an example.

## 2.2 Emissions and model experiments

Baseline emissions for 2015 were from the ECLIPSE V6 (Evaluating the CLimate and Air Quality ImPacts of Short-livEd Pollutant) inventory, available at https://previous.iiasa.ac.at/web/home/research/researchPrograms/air/ECLIPSEv6b.html (last access: 8 August 2022). Monthly emissions profiles derived from EDGAR (Emission Database for Global Atmospheric Research, v4.3.2 datasets, available at https://edgar.jrc.ec.europa.eu/dataset_temp_profile) time series (Crippa et al., 2020) were applied to the ECLIPSE annual emissions of $SO_2$, $NO_2$, $NH_3$, CO, $CH_4$, NMVOC, primary $PM_{2.5}$ and coarse particles. Detailed implementation procedures including the re-assignment of ECLIPSE emissions sectors to EMEP sectors and the calculation of temporal profiles at a given country for a given pollutant are described in Ge et al. (2021b).

A baseline simulation and a set of 8 sensitivity experiments were conducted for emissions and meteorology for 2015. Limited by available computational resources and storage space and taking the achievability of real-world emissions controls into account, the model experiments applied 20% and 40% reductions to global anthropogenic emissions of $NH_3$, $NO_x$, $SO_x$ from all sectors both individually and collectively (i.e., reductions applied to all 3 species simultaneously). All other emissions, including natural emissions such as dimethyl sulfide from oceans, lightning $NO_x$, and soil $NO_x$, were left unchanged.

The sensitivity ($Sensitivity_i$) of the concentration/deposition of a species $i$ is calculated as the absolute difference between the value in baseline ($Baseline_i$) and in an emission reduction scenario ($Scenario_i$). Taking $NH_3$ concentration as an example:

$$Sensitivity_{NH_3}(\mu g\ m^{-3}) = Baseline_{NH_3} - Scenario_{NH_3}$$

For the relative sensitivity ($Relative\ Sensitivity_i$):

$$Relative\ Sensitivity_{NH_3}(\%) = \frac{Sensitivity_{NH_3}}{Baseline_{NH_3}} \times 100\%$$

The sensitivities of different species are calculated for all emission reduction scenarios. The $PM_{2.5}$ sensitivities derived from individual reductions in emissions of $NH_3$, $NO_x$, or $SO_x$ are used to define the sensitivity regimes for different regions in Sect. 3.2. For each model grid, the regime is decided by the precursor that yields the greatest decrease in grid $PM_{2.5}$ concentration: $NH_3$ sensitive, $NO_x$ sensitive, or $SO_x$ sensitive.

## 2.3 Definition of world regions

We compared the sensitivities to the emissions reductions of $PM_{2.5}$, $N_r$ and $S_r$ species concentrations and depositions in the four world regions of East Asia, South Asia, Euro_Medi, and North America defined in Fig. 1 (and listed in Table S1). These are based on regions used by the Intergovernmental Panel on Climate Change and as rationalised in Iturbide et al. (2020).

All four regions are densely populated and have high $N_r$ and $S_r$ pollution. Besides, due to limitations in the number of publicly available measurements, our model outputs are evaluated against measurements in East Asia, Europe, and North America, and therefore we have greater confidence in sensitivity results in these three regions. South Asia is chosen because of its extreme ammonia-richness, as revealed by Ge et al. (2022), which makes it an interesting comparison with other regions.

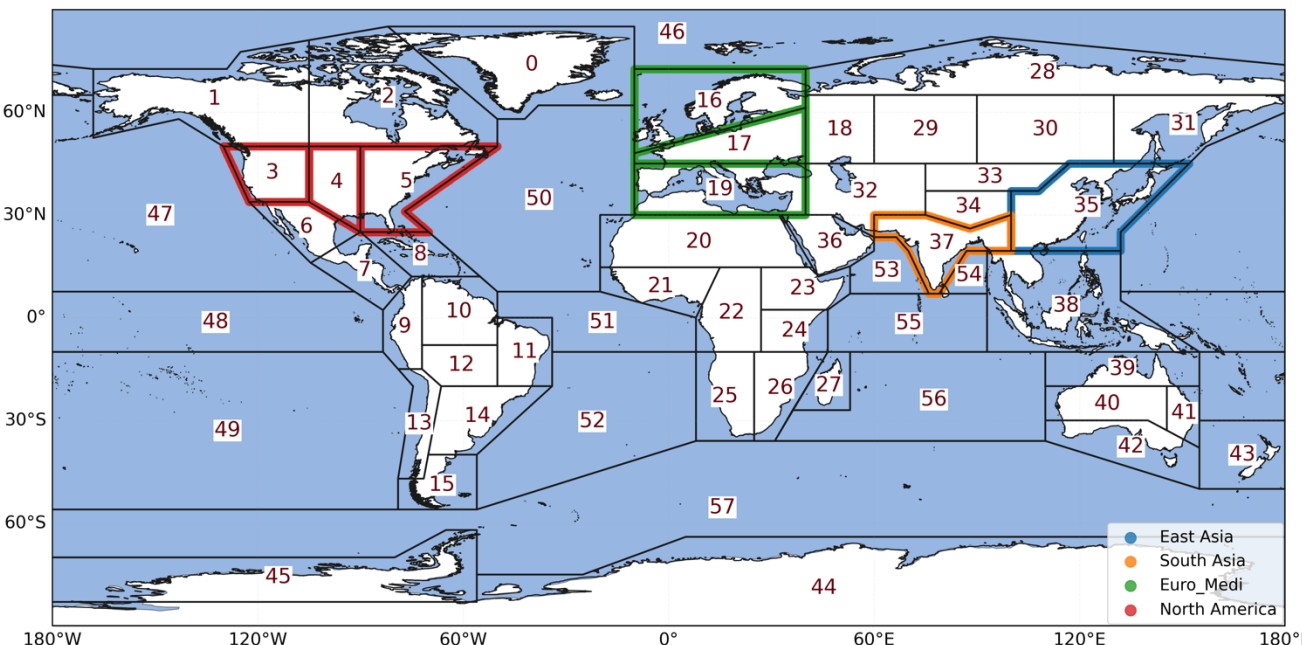

**Figure 1: The boundaries of the 4 world regions used in this study, which are based on the IPCC reference regions described in Iturbide et al. (2020).**

## 3 Results

### 3.1 Sensitivities of $N_r$ and $S_r$ gas and aerosol concentrations

The simulated global baseline 2015 annual mean surface concentrations of $N_r$ and $S_r$ have been discussed in detail in Ge et al. (2022). Here we analyse the sensitivities of the modelled surface concentrations to SIA precursor emissions reductions for RDN, OXN and OXS components. Sensitivities differ according to consideration of primary or secondary components and show great geographical variation.

### 3.1.1 RDN

Figure 2 shows the spatial variations in the sensitivities of $NH_3$ and $NH_4^+$ annual mean surface concentrations to 20% and 40% emissions reductions in $NH_3$, $NO_x$, and $SO_x$ individually, and collectively. Regional average sensitivities in East Asia, South Asia, Euro_Medi, and North America are summarised in Fig. 3 and Table S2. Steeper gradients in Fig. 3 correspond to greater concentration changes (sensitivities).

From these figures, it is clear that whilst reducing emissions of $NH_3$ (and all 3 precursors together) decrease $NH_3$ concentrations efficiently, reducing emissions of $NO_x$ or $SO_x$ lead to increases in $NH_3$ concentrations, particularly over densely populated areas. The maximum reduction in model grid $NH_3$ concentration across all scenarios reaches 16.6 µg m⁻³ (44% change relative to baseline; expressed similarly hereafter) and occurs over East Asia in response to a 40% reduction in $NH_3$ emissions. The largest increase in $NH_3$ concentration (1.51 µg m⁻³, 22%) arises in Southeast Asia under the 40% $SO_x$ emissions reduction scenario. Lower $NO_x$ and $SO_x$ emissions decrease the concentration of acidic species available to react with $NH_3$ to form $NH_4^+$ aerosol, so more of the emitted $NH_3$ stays in the gas phase. Other studies have likewise shown that reductions in

SO$_2$ or NO$_x$ emissions are an important contributor to the growth in tropospheric NH$_3$ concentrations globally and regionally (Saylor et al., 2015; Warner et al., 2017; Liu et al., 2018; Yu et al., 2018).

In East Asia and North America, NH$_3$ concentrations increase similarly for either NO$_x$ or SO$_x$ reductions (Fig. 3), but in South Asia and Euro_Medi, NH$_3$ concentrations increase more with SO$_x$ reductions than with NO$_x$ reductions, which reflects the larger contribution of (NH$_4$)$_2$SO$_4$ than NH$_4$NO$_3$ to SIA in the latter two regions. However, the increase in NH$_3$ concentrations is relatively small compared to the extent of NO$_x$ and SO$_x$ emissions reductions: 40% reductions in emissions of NO$_x$ or SO$_x$ only increase NH$_3$ concentrations in the 4 regions by 2-6% or 6-9%, respectively (Fig. 3, Table S2). The globally averaged increases in NH$_3$ concentrations for 40% reductions in NO$_x$ or SO$_x$ emissions are 3% and 9%, respectively. Nevertheless, the NH$_3$ concentration decrease resulting from reductions in NH$_3$ emissions is offset by simultaneous effects of NO$_x$ and SO$_x$ emissions reductions when all 3 precursors are reduced together, as the sensitivities of regional average NH$_3$ concentrations to 40% reductions in all 3 precursor emissions (38-39% across the four regions) are smaller than their sensitivities to 40% reductions in NH$_3$ emissions on its own (47-49%). It is also noteworthy that the sensitivities of regional average NH$_3$ concentrations are essentially linear through 20% and 40% emissions reductions, irrespective of precursor, although the sensitivities are different between regions.

In contrast to NH$_3$, concentrations of NH$_4^+$ always decrease when an SIA precursor emission is reduced. Figure 2 shows that NH$_4^+$ concentrations in the most densely populated continents (e.g., eastern China, India, Europe, eastern America) respond strongly to emissions reductions in each SIA precursor, whilst they only respond to SO$_x$ emissions reductions over oceans. This is related to the production of marine sulfate aerosol from dimethyl sulfide (DMS) and the lack of significant oceanic NO$_x$ emissions sources, which means only (NH$_4$)$_2$SO$_4$ formation is important in marine SIA chemistry.

In addition, the impacts of NH$_3$ and SO$_x$ emissions reductions on NH$_4^+$ concentrations over North Africa are significantly greater than from NO$_x$ emissions reductions, indicating a dominance of (NH$_4$)$_2$SO$_4$ within SIA in this region. This is consistent with the results reported by Ge et al. (2022). They showed that large areas in North Africa are characterised by the SO$_4^{2-}$-rich chemical domain for SIA formation, which means that NH$_3$ is predominantly taken up by SO$_4^{2-}$, leaving no free NH$_3$ to react with HNO$_3$ to form NH$_4$NO$_3$. Given that emissions reductions in all three precursors individually lead to reductions in NH$_4^+$ concentrations, it is not surprising that the greatest simulated NH$_4^+$ reduction (5.87 μg m$^{-3}$ (43%) in East Asia) arises for the scenario with 40% reductions in all 3 precursors collectively.

Figure 3 also shows that NH$_4^+$ sensitivities are essentially linear for emissions reductions to 40%, although responses again vary slightly with region. Among individual precursor reduction scenarios, regional average NH$_4^+$ concentrations in East Asia and Euro_Medi are most sensitive to SO$_x$ emissions reductions and least sensitive to NO$_x$ reductions, while NH$_4^+$ concentrations in North America are most sensitive to SO$_x$ reductions and least sensitive to NH$_3$ reductions. In South Asia, NH$_4^+$ is characterised by strong sensitivity to SO$_x$ emissions reductions but only relatively small sensitivities to NO$_x$ and NH$_3$ emissions reductions. In the scenario of all 3 species reductions, all regions show relative sensitivities close to the one-to-one line. Another important observation from Fig. 3 is that reductions in NH$_4^+$ (4-18%) in response to a 40% NH$_3$ emissions reduction are much smaller than reductions of NH$_3$ concentrations (47-49%) in these regions, which reflects the fact that these regions are so ammonia-rich that reducing NH$_3$ emissions only has limited effects on NH$_4^+$ concentrations.

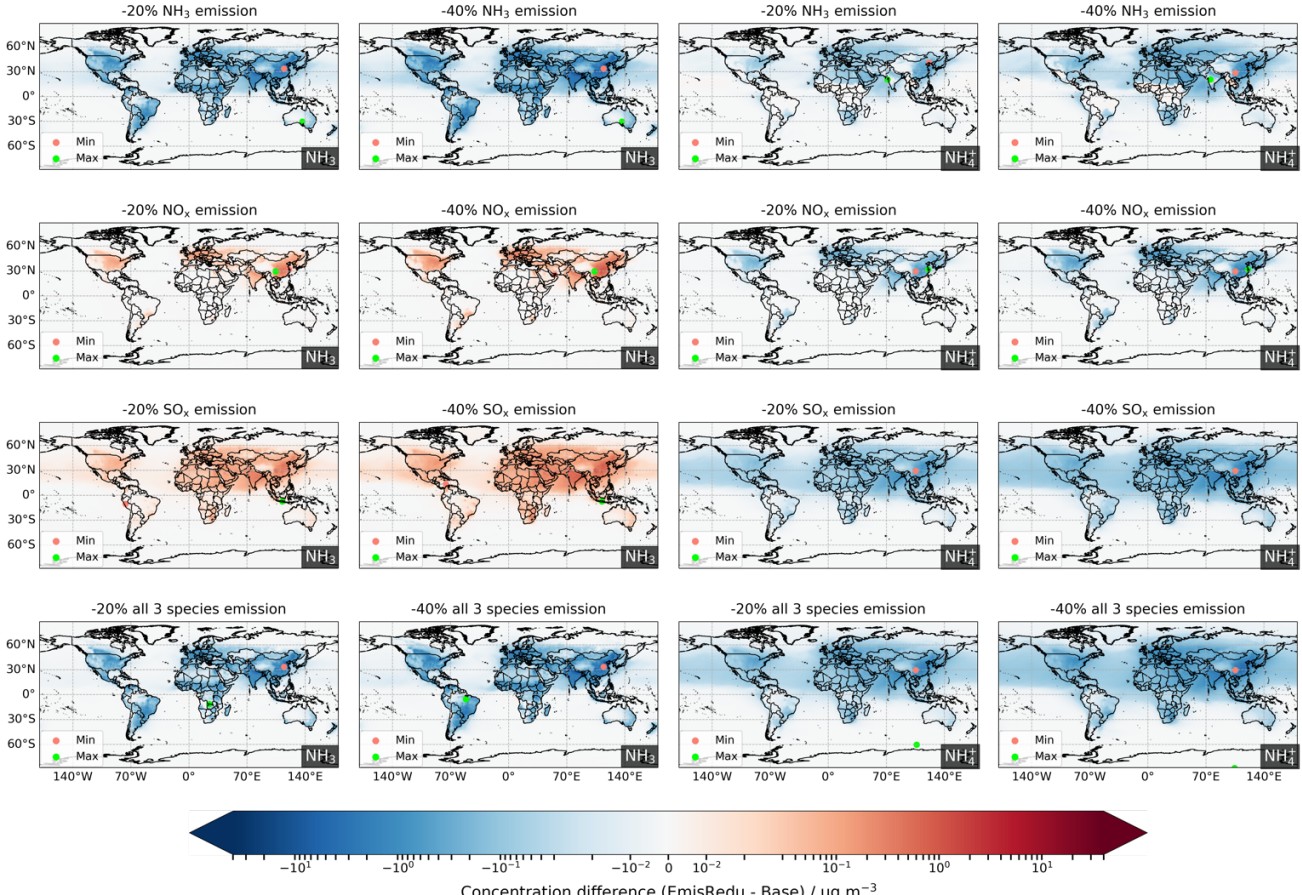

**Figure 2: Changes in NH₃ and NH₄⁺ annual surface concentrations for 20% and 40% emissions reductions in NH₃, NOₓ, and SOₓ individually and collectively. Red and green dots in each map locate the minimum and maximum difference, respectively.**

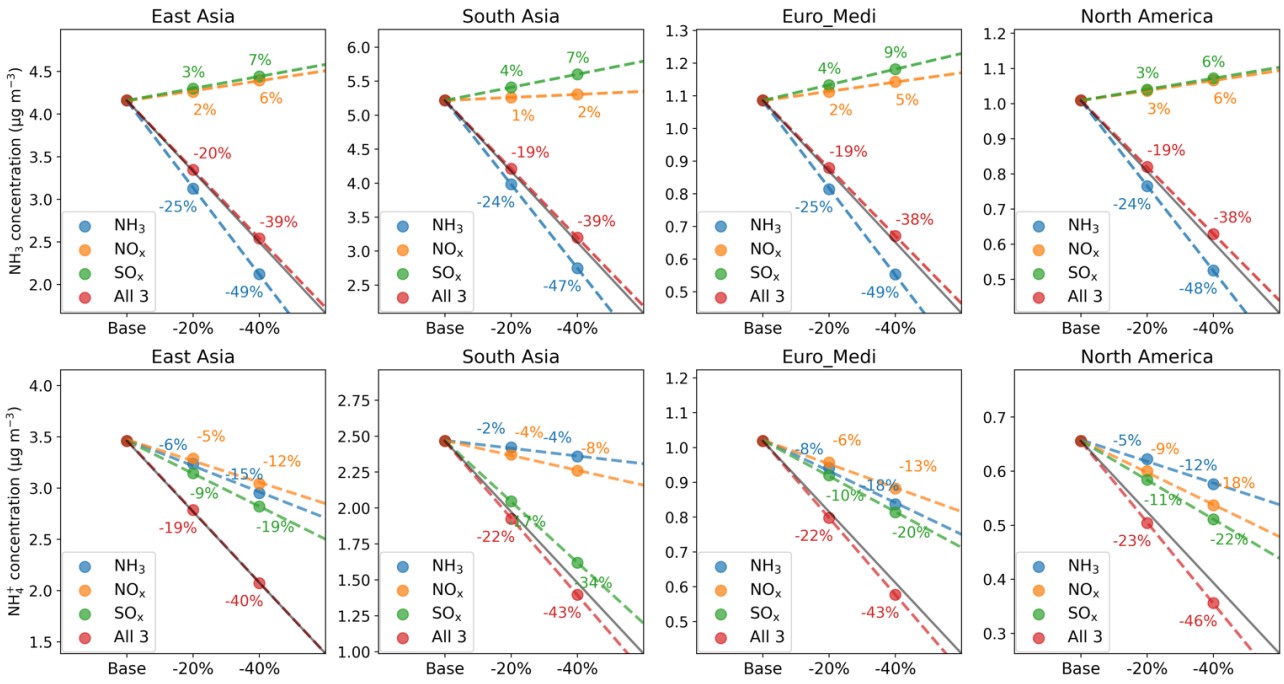

**Figure 3: The absolute and relative sensitivities of regionally-averaged annual mean surface concentrations of NH₃ (upper row) and NH₄⁺ (lower row) to 20% and 40% emissions reductions in NH₃ (blue), NOₓ (orange) and SOₓ (green) individually, and collectively (red), for the four regions defined in Fig. 1. The solid grey line in each panel illustrates the one-to-one relative response to emissions reductions, whilst the coloured dashed lines are the linear regressions through each set of three model simulations and illustrate the actual responses to emissions reductions of a given precursor. The numbers show the corresponding relative responses to each emissions reduction (with respect to baseline).**

**3.1.2 OXN**

Figure 4 shows the spatial variations in the sensitivities of $NO_x$ and fine nitrate annual mean surface concentrations to 20% and 40% emissions reductions in $NH_3$, $NO_x$, and $SO_x$ individually, and collectively. Regional average sensitivities in East Asia, South Asia, Euro_Medi, and North America are summarised in Fig. 5 and Table S3. Equivalent global maps and regional sensitivity plots for the responses of $HNO_3$ and coarse nitrate to the same emissions reductions are presented in Figs. S2 and S3 and Table S3.

In contrast to surface $NH_3$, whose concentrations are sensitive to reductions in emissions of each of $NH_3$, $NO_x$ and $SO_x$, surface $NO_x$ concentrations only respond to $NO_x$ emissions reductions and have negligible sensitivity to $NH_3$ and $SO_x$ emissions reductions. The 20% and 40% reductions in $NO_x$ emissions yield a global maximum of 25.6 μg m$^{-3}$ (22%) and 51.3 μg m$^{-3}$ (44%) reductions in surface $NO_x$ concentrations over East Asia. The globally averaged reductions in $NO_x$ concentrations for 20% and 40% reductions in $NO_x$ emissions are 15% and 30% respectively (same values for simultaneous reductions in all 3 precursors), whereas the sensitivities for $NH_3$ and $SO_x$ emissions reduction are all 0% (Table S3).

Regionally (Fig. 5), East Asia shows the largest decreases in $NO_x$ concentrations (45%) in response to 40% $NO_x$ emissions reductions, followed by Euro_Medi and North America (36-38%), and South Asia (23%). These regional differences in sensitivities to $NO_x$ emissions are due to regional differences in oxidation chemistry climate. Figure S4 shows the changes in annual mean surface concentrations of $O_3$ for the 8 emissions reduction scenarios. Concentrations of $O_3$ in eastern China, western and central Europe, and north-eastern US increase as $NO_x$ emissions reduce, while $O_3$ in the rest of the world decreases as $NO_x$ emissions reduce. In East Asia, the increased oxidant levels enhance $NO_x$ chemical removal and results in a greater than one-to-one relative decrease in $NO_x$ concentrations with $NO_x$ emissions reductions. Consequently, the decrease in fine nitrate in East Asia is offset by enhanced chemical production, which leads to a lower than one-to-one sensitivity (e.g., a 40% reduction in $NO_x$ emissions gives a 33% decrease in fine $NO_3^-$). In contrast, decreased oxidant levels in South Asia decrease the oxidation of $NO_x$, which partially offsets the decrease in $NO_x$ concentrations induced by emissions reductions, causing a more efficient reduction in fine $NO_3^-$ concentrations than in $NO_x$ in this region. The variation in regional atmospheric oxidising capacity also alters the $SO_4^{2-}$ formation processes; discussions of this are presented in Sect.3.1.3 and Sect. 4. The situation is more complex for Euro_Medi and North America as these regions include both positive and negative changes in $O_3$ concentrations with $NO_x$ emissions reductions (Fig. S4) and they are not as $NO_x$-rich as East Asia and South Asia. The effects of changes in oxidant levels on $NO_x$ and fine $NO_3^-$ concentrations are therefore more localised and less apparent in regional averages. Clappier et al. (2021) reported this effect to be most distinct in the Po basin (Italy), western Germany, and Netherlands in Europe, whilst for the United States, Tsimpidi et al. (2008) showed it only becomes pronounced in the northeast, both of which are consistent with our results.

For secondary OXN species, the sensitivities of $HNO_3$ and fine and coarse $NO_3^-$ to individual reductions in emissions of $NH_3$, $NO_x$ and $SO_x$ are closely associated with SIA formation chemistry. The principal observation from Fig. 4 is that concentrations of fine $NO_3^-$ in all four regions decrease with reduced $NH_3$ and $NO_x$ emissions but increase with reduced $SO_x$ emissions. This is because $H_2SO_4$ and $HNO_3$ compete in their reactions with $NH_3$, and $(NH_4)_2SO_4$ is formed preferably over $NH_4NO_3$. Reductions in $NH_3$ emissions cause the equilibrium between $HNO_3$ and $NH_3$ to shift away from $NH_4NO_3$ production and therefore free more $HNO_3$ molecules. As a result, more $HNO_3$ is available to produce coarse nitrate aerosol, leading to a decrease in fine $NO_3^-$ but an increase in coarse $NO_3^-$ concentrations (Fig. S2). Reductions in $NO_x$ emissions decrease $HNO_3$ and fine and coarse $NO_3^-$ concentrations globally. Although the increased oxidant levels that arise in some regions following $NO_x$ emissions reductions enhance the chemical formation of these secondary species (as discussed above), $NO_x$ emissions reductions of 20% and 40% are substantial enough to mean that the lower availability of $NO_x$ to form $NO_3^-$ dominates the impact on $NO_3^-$ concentrations compared with the enhancement in oxidising capacity. Reduced $SO_x$ emissions leave more $NH_3$ to equilibrate with $HNO_3$ to form $NH_4NO_3$, leading to an increase in fine $NO_3^-$ concentrations but to a decrease in coarse $NO_3^-$ concentrations as the former takes more $HNO_3$. It is notable that the increase in fine $NO_3^-$ concentrations is relatively small

compared to the extent of SO$_x$ emissions reductions. For example, the maximum increase in fine NO$_3^-$ resulting from 40% reductions in SO$_x$ emissions is 1.71 µg m$^{-3}$ (16%), in East Asia. The regional average increases in fine NO$_3^-$ concentrations for 40% SO$_x$ emissions reductions are 8% in East Asia, South Asia, and Euro_Medi, and 4% in North America.

The differences in regional average sensitivities of HNO$_3$ and fine and coarse NO$_3^-$ are highlighted more clearly in Fig. 5 and Fig. S2. Fine NO$_3^-$ in East Asia is equally sensitive to NO$_x$ and NH$_3$ emissions reductions (33% and 32% decreases for 40% NO$_x$ and NH$_3$ emissions reductions respectively), while it is more sensitive to NO$_x$ emissions reductions than to NH$_3$ emissions reductions in South Asia (45% and 39%), Euro_Medi (41% and 33%), and North America (42% and 26%). In terms of absolute concentration changes, the reductions in fine NO$_3^-$ over East Asia in response to 40% NH$_3$ and NO$_x$ emissions reductions (1.62 -1.65 µg m$^{-3}$) are more than 3 times larger than reductions in other regions (0.23 - 0.47 µg m$^{-3}$). On the other hand, if NH$_3$ emissions are reduced then the increases in HNO$_3$ and coarse NO$_3^-$ concentrations in East Asia (15% increases for 40% NH$_3$ reductions) are much larger than the increases in the other three regions (2-6%). All these differences between East Asia and the other three regions reflect the larger abundance of NH$_4$NO$_3$ in SIA over East Asia. This is demonstrated in Fig. S8 which shows that the contribution of fine NO$_3^-$ to PM$_{2.5}$ in the baseline is greatest in East Asia (19%, 5.21 µg m$^{-3}$), followed by Euro_Medi (12%, 1.22 µg m$^{-3}$), North America (11%, 0.86 µg m$^{-3}$), and South Asia (3%, 0.93 µg m$^{-3}$). Detailed discussion on regional SIA composition is presented in Sect. 3.2.

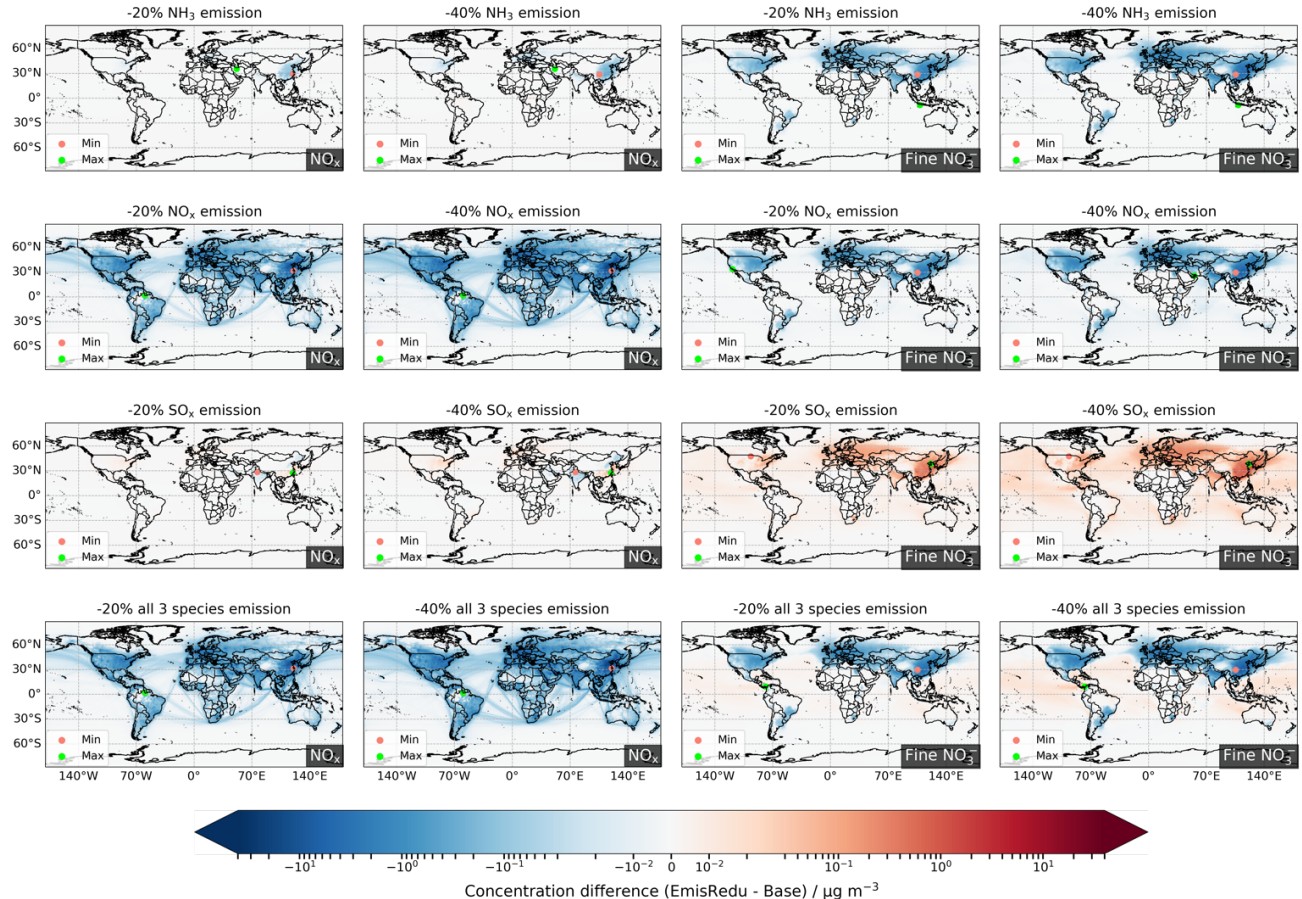

**Figure 4: Changes in NO$_x$ and fine NO$_3^-$ annual surface concentrations for 20% and 40% emissions reductions in NH$_3$, NO$_x$, and SO$_x$ individually and collectively. Red and green dots in each map locate the minimum and maximum difference, respectively.**

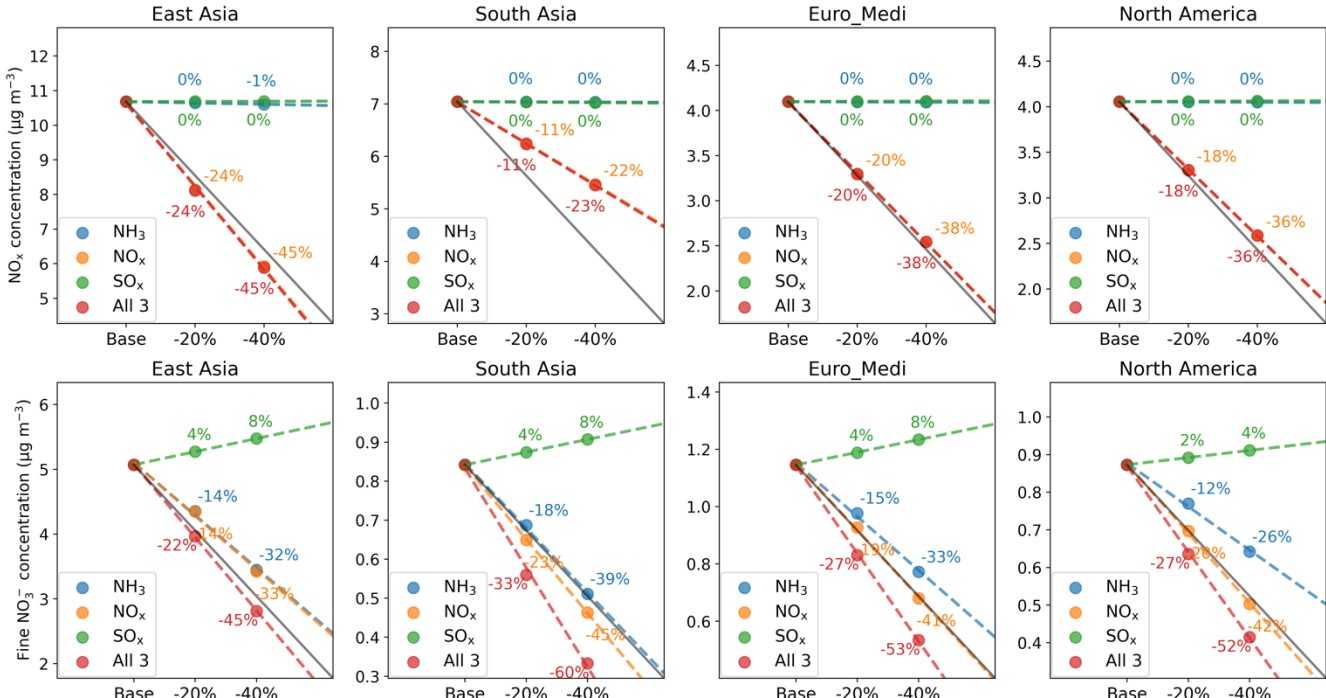

**Figure 5:** The absolute and relative sensitivities of regionally-averaged annual mean surface concentrations of $NO_X$ (upper row) and fine $NO_3^-$ (lower row) to 20% and 40% emissions reductions in $NH_3$ (blue), $NO_x$ (orange) and $SO_x$ (green) individually, and collectively (red), for the four regions defined in Fig. 1. The solid grey line in each panel illustrates the one-to-one relative response to emissions reductions, whilst the coloured dashed lines are the linear regressions through each set of three model simulations and illustrate the actual responses to emissions reductions of a given precursor. The numbers show the corresponding relative responses to each emissions reduction (with respect to baseline).

### 3.1.3 OXS

Figure 6 shows the global variation in the sensitivities of $SO_2$ and $SO_4^{2-}$ annual mean surface concentrations to 20% and 40% emissions reductions in $NH_3$, $NO_x$, and $SO_x$ individually, and collectively. Figure 7 and Table S4 summarise the sensitivities of the regionally averaged $SO_2$ and $SO_4^{2-}$ concentrations to the emissions reductions for the four regional domains. Concentrations of $SO_2$ increase in response to reduced $NH_3$ emissions particularly over East Asia, South Asia, Europe, and North America. The largest increases in $SO_2$ resulting from 20% and 40% $NH_3$ emissions reductions are respectively in Southeast Asia (3.62 μg $m^{-3}$, 9%) and East Asia (8.11 μg $m^{-3}$, 20%) (Fig. 6). The response of secondary $SO_4^{2-}$ concentrations to $NH_3$ emissions reductions varies substantially across the world. In north-eastern China and Europe, $SO_4^{2-}$ concentrations increase, whilst in southern China, India, and United States they decrease. However, the magnitudes of $SO_4^{2-}$ concentration changes are much smaller than for $SO_2$. The maximum increases in $SO_4^{2-}$ concentrations (located in southern China) are only 0.37 μg $m^{-3}$ (2%) and 0.89 μg $m^{-3}$ (5%) for 20% and 40% $NH_3$ emissions reductions, respectively. The maximum decreases in $SO_4^{2-}$ concentrations, in north-eastern China, are 0.62 μg $m^{-3}$ (6%) and 1.30 μg $m^{-3}$ (12%), respectively. Regionally averaged, East Asia exhibits the largest increase in $SO_2$ for 40% $NH_3$ emissions reductions (16%) (Fig. 7) followed by North America (14%), South Asia (14%), and Euro_Medi (10%), whereas increases in regional average $SO_4^{2-}$ concentrations are only in the range 0-2%.

Figures S5 and 13 show the global variation in the sensitivities to precursor emissions reductions of total deposition of $SO_2$, and of the wet and dry deposition of all OXS components, respectively. When $NH_3$ emissions are reduced, $SO_2$ total deposition decreases over all of East Asia, South Asia, Europe, and North America (Fig. S5); Fig. 13 shows the decrease is driven by reduced $SO_2$ dry deposition. Several studies have shown that the non-stomatal canopy uptake resistance of $SO_2$ (the inverse of the $SO_2$ dry deposition velocity) is positively correlated to the molar 'acidity ratio' $a_{SN} = \frac{[SO_2]}{[NH_3]}$ (Smith et al., 2000; Erisman et al., 2001; Fowler et al., 2009; Massad et al., 2010), a process that is included in the EMEP MSC-W model (Simpson

et al., 2012). Reduced $NH_3$ concentrations therefore increase the acidity ratio and hence decrease the rate of $SO_2$ dry deposition and increase the $SO_2$ surface concentrations in those regions where this effect is significant. The $SO_4^{2-}$ responses to $NH_3$ emissions reductions are related to changes in atmospheric acidity as well. The aqueous-phase oxidation of $SO_2$ by $O_3$, which is one of the major pathways for $SO_4^{2-}$ production, is significantly pH dependent. In general, the oxidation rate decreases with decreased pH (Penkett et al., 1979; Maahs, 1983; Liang and Jacobson, 1999; Hattori et al., 2021), a process that is incorporated into the EMEP MSC-W model (Simpson et al., 2012). As pointed out by Ge et al. (2022), Europe and north-eastern China are much less ammonia-rich than India, which means that 20% and 40% $NH_3$ emissions reductions are substantial enough to decrease the pH in the former two regions, leading to decreases in $SO_4^{2-}$ production. However, given that $SO_2$ levels increase and that there are still other effective oxidation pathways (e.g., OH, $H_2O_2$) which are independent of pH (McArdle and Hoffmann, 1983; Hoffmann, 1986; Seinfeld and Pandis, 2016), the decreases in $SO_4^{2-}$ concentrations in Europe and north-eastern China due to $NH_3$ emissions reductions are very small anyway. In contrast, since India is so ammonia-rich, even 40% reductions in $NH_3$ emissions do not significantly alter the pH in this region. As a result, $SO_4^{2-}$ concentrations in India increase slightly due to the higher availability of its precursor $SO_2$.

The impacts of $NO_x$ emissions reductions on $SO_2$ concentrations show inverse trends in different regions (Fig 7), which reflects regional differences in atmospheric oxidation chemistry. The decreased $SO_2$ concentrations (maximum reduction: 1.72 $\mu g\ m^{-3}$, 6%) in eastern China, Europe, and north-eastern United States (Fig. 6) can be explained by the enhanced $O_3$ concentrations in these regions arising from the reduced $NO + O_3$ reaction in these high $NO_x$ regions (Fig. S4). As a result, $SO_2$ is more readily oxidised to $SO_4^{2-}$, leading to increased $SO_4^{2-}$ concentrations (maximum increase for 40% $NO_x$ emissions reductions: 1.59 $\mu g\ m^{-3}$ (10%)) in these regions. This positive response of $SO_4^{2-}$ to $NO_x$ emissions reductions is also reported in regional studies (Botha et al., 1994; Li et al., 2006; Sheng et al., 2018; Fang et al., 2019; Ge et al., 2021a). In contrast, India, north-eastern Africa, and southern Africa show increased $SO_2$ (maximum increase: 0.29 $\mu g\ m^{-3}$, 1%) but decreased $SO_4^{2-}$ concentrations (maximum decrease: 0.73 $\mu g\ m^{-3}$, 9%) as $NO_x$ emissions reduce, which can be explained by the parallel decrease in $O_3$ concentrations in these regions (Fig. S4). However, these concentration changes are very localised and, from a regional average perspective, are relatively small compared to the extent of emissions reductions applied. For example, in East Asia, the region with the largest response, there is only a 5% decrease in regional average $SO_2$ concentration (Fig. 7), and 3% increase in $SO_4^{2-}$ concentration, for a 40% reduction in $NO_x$ emissions. For other regions, the $SO_2$ and $SO_4^{2-}$ regional average concentration changes are even smaller (from -4% to 2%). The global average sensitivities of $SO_2$ and $SO_4^{2-}$ annual mean concentrations to 20% and 40% $NO_x$ emissions reductions are only in the range 0-2% (Table S4).

Under reductions of $SO_x$ emissions (and of all 3 precursors together), both $SO_2$ and $SO_4^{2-}$ show almost one-to-one reductions, indicating that $SO_x$ emissions reductions are crucial for reducing both primary and secondary OXS pollutants and, in the case of reductions of all 3 precursors simultaneously, readily sufficient to dominate over any tendency for $NH_3$ and $NO_x$ emissions reductions to increase OXS species concentrations. A 40% reduction in $SO_x$ emissions leads to a maximum $SO_2$ decrease of 39.8 $\mu g\ m^{-3}$ (40%), in northern Russia, and a maximum $SO_4^{2-}$ decrease of 7.70 $\mu g\ m^{-3}$ (39%), in south-eastern China (Fig. 6). For the four regions, average $SO_2$ concentrations decrease by 22-24% and 42-45%, and $SO_4^{2-}$ concentrations decrease by 17-19% and 34-38%, in response to 20% and 40% $SO_x$ emissions reductions respectively (Fig. 7). For the 20% and 40% reductions in all 3 precursors together, regionally averaged $SO_2$ decrease by 19-20% and 38-41% respectively, and regionally averaged $SO_4^{2-}$ decrease by 17-20% and 35-40% respectively.

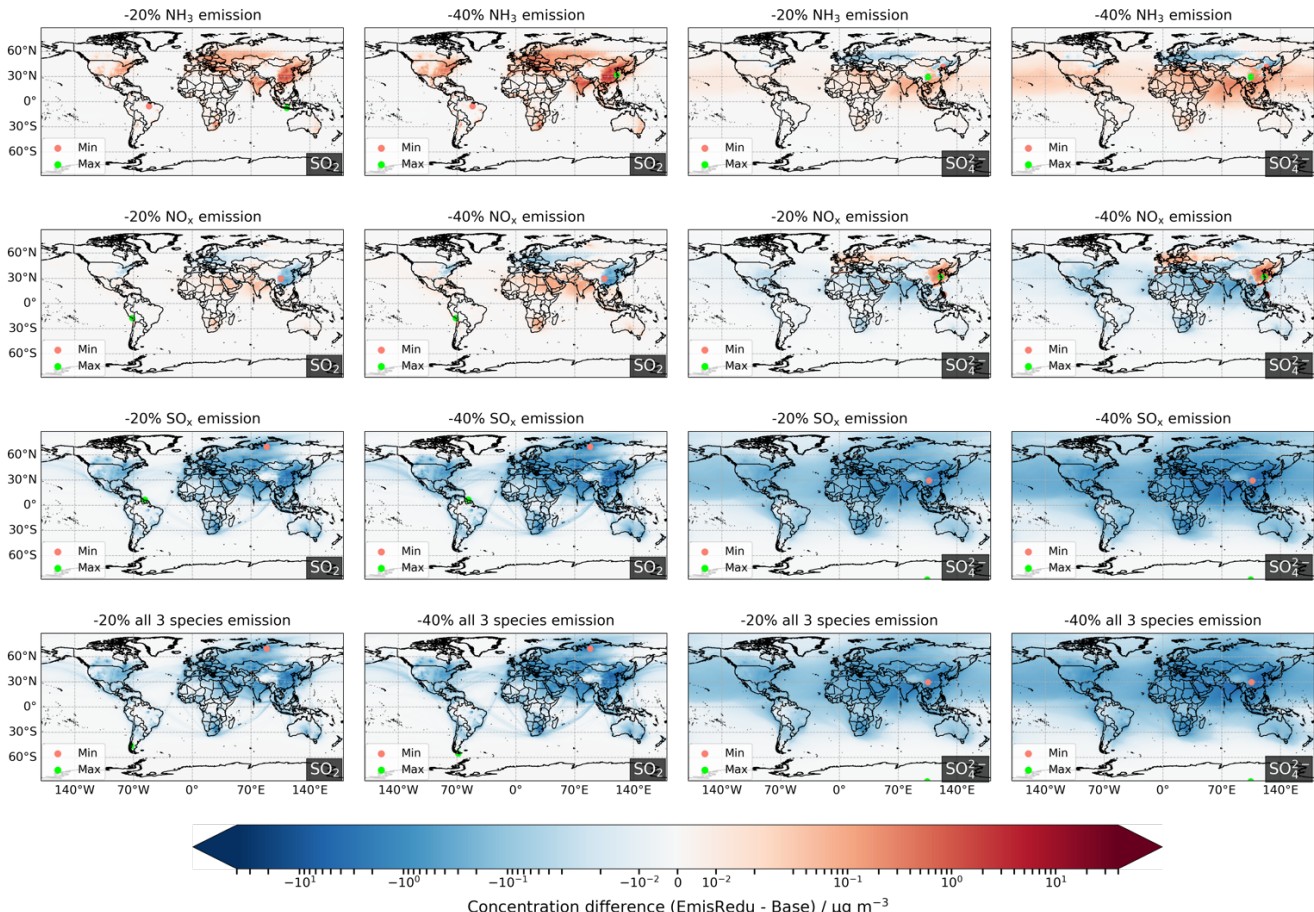

**Figure 6: Changes in SO₂ and SO₄²⁻ annual surface concentrations for 20% and 40% emissions reductions in NH₃, NOₓ, and SOₓ individually and collectively. Red and green dots in each map locate the minimum and maximum difference, respectively.**

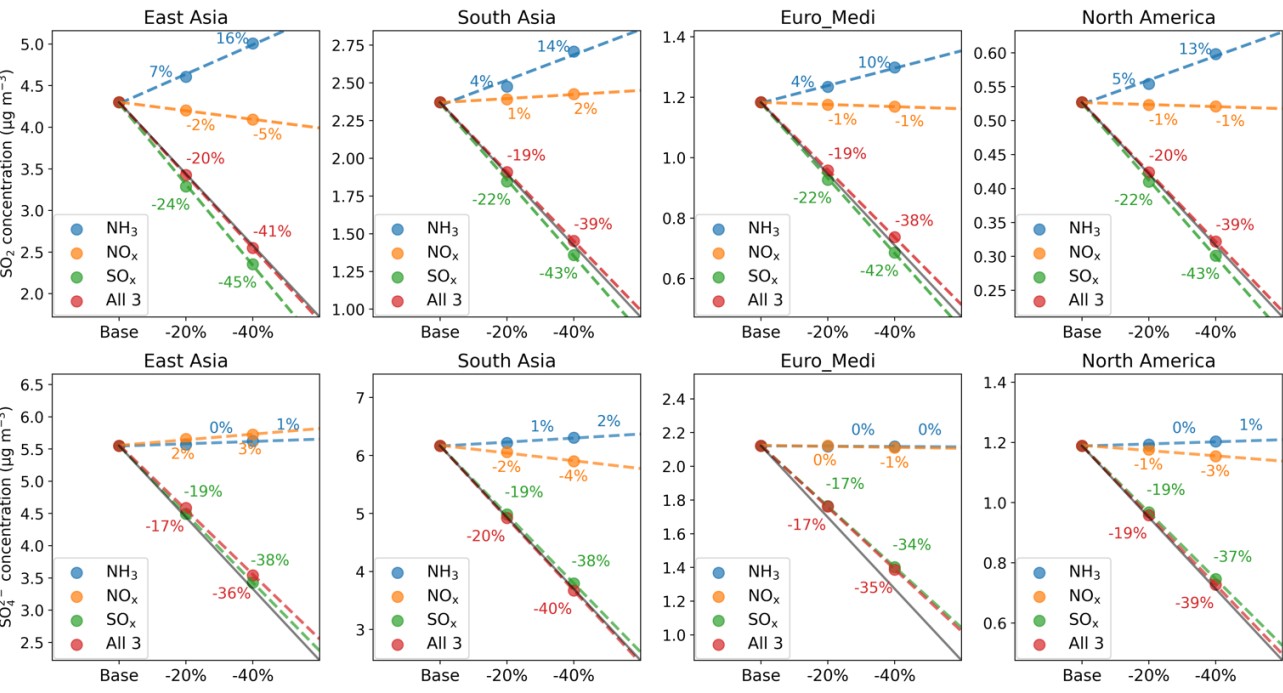

**Figure 7: The absolute and relative sensitivities of regionally-averaged annual mean surface concentrations of SO₂ (upper row) and SO₄²⁻ (lower row) to 20% and 40% emissions reductions in NH₃ (blue), NOₓ (orange) and SOₓ (green) individually, and collectively (red), for the four regions defined in Fig. 1. The solid grey line in each panel illustrates the one-to-one relative response to emissions reductions, whilst the coloured dashed lines are the linear regressions through each set of three model simulations and illustrate the actual responses to emissions reductions of a given precursor. The numbers show the corresponding relative responses to each emissions reduction (with respect to baseline).**

## 3.2 Sensitivity of PM₂.₅ concentrations

Figure 8 shows the global distribution of the dominant sensitivity of $PM_{2.5}$ towards a 40% reduction in $NH_3$, $NO_x$, or $SO_x$ emissions individually. An annual mean $PM_{2.5}$ concentration threshold of 5 µg m⁻³ has been applied in order to focus attention on more polluted areas where $PM_{2.5}$ concentrations in the baseline simulation are above the latest WHO $PM_{2.5}$ air quality guideline (AQG) (WHO, 2021). Section S2 and Fig. S7 in the Supplement present seasonal variations in global $PM_{2.5}$ sensitivity regimes.

The principal observation from Fig. 8 is that the sensitivity of $PM_{2.5}$ to reductions in emissions of individual precursors is highly geographically variable. $SO_x$-sensitive regimes are found in Southeast Asia, South Asia, Africa, and Central America. $NO_x$-sensitive regimes are observed in south-eastern China, France, Germany, most eastern European countries, central and eastern United States, and northern and central parts of South America. Only a few small regions are $NH_3$ sensitive: these include eastern coastal areas around China, the UK and its surrounding seas, southern Scandinavia, and western Russia. The difference in $PM_{2.5}$ sensitivity between northern Europe and the rest of Europe demonstrates that $NH_3$ has become the limiting factor for SIA formation in northern Europe. This greater leverage of $NH_3$ emissions on $PM_{2.5}$ mitigation in this region is due to the effective emissions controls on all SIA precursor emissions here (see also Sect. 4 discussion) (Tørseth et al., 2012; AQEG, 2015; Vieno et al., 2016; Ciarelli et al., 2019; Theobald et al., 2019). In contrast, South Asia is so ammonia-rich that reducing $NH_3$ concentrations has little impact on $PM_{2.5}$ (Ge et al., 2022). The situation in northern Europe exemplifies what to expect in terms of future policy making for the rest of the world. Furthermore, many marine areas are characterised as $SO_x$ sensitive but for a different reason than the $SO_x$-sensitive regime in South Asia. In the marine areas, fine nitrate and ammonium aerosols are relatively small compared to sulfate aerosols, therefore reductions in $NO_x$ and $NH_3$ emissions hardly affect SIA formation. In fact, sulfate aerosol derived from oceanic emissions of DMS rather than from anthropogenic emissions is the major contributor to marine $PM_{2.5}$ (Quinn and Bates, 2011; Hoffmann et al., 2016; Novak et al., 2022).

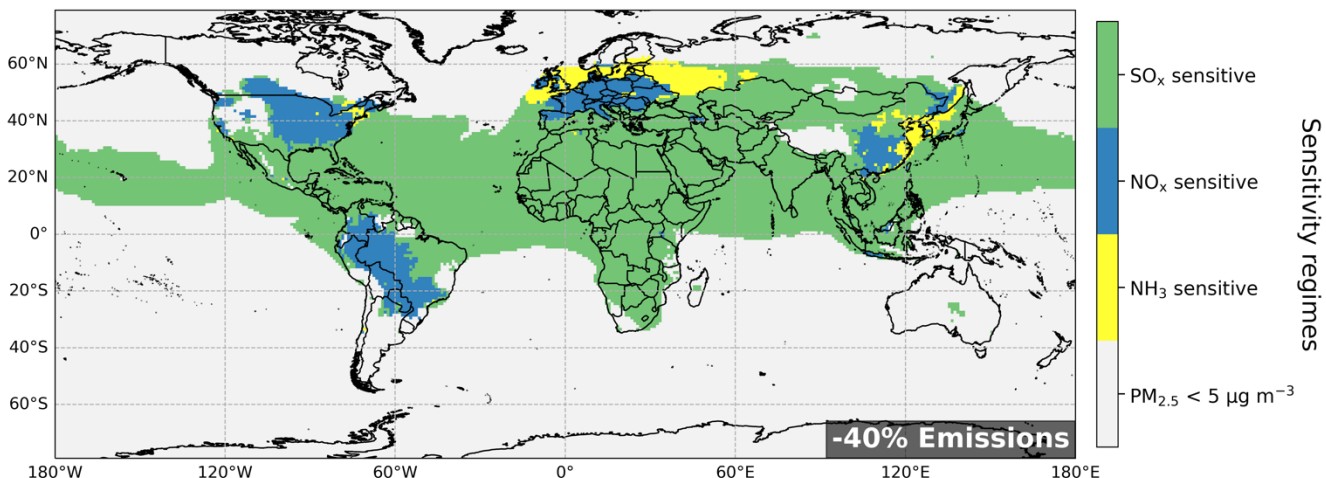

**Figure 8: Spatial variation in sensitivity regime of PM₂.₅ mitigation based on data from 40% individual reductions in emissions of NH₃, NOₓ, or SOₓ. The regime is defined according to the precursor that yields the greatest decreases in grid PM₂.₅ concentration: NH₃ sensitive (yellow), NOₓ sensitive (blue), SOₓ sensitive (green). Model grids with baseline annual mean PM₂.₅ concentrations <5 µg m⁻³ are masked out.**

Detailed examination of the magnitudes of the $PM_{2.5}$ sensitivities in each location to each precursor (rather than just their ranking) reveals more complicated regional characteristics. Figure 9 shows the spatial variabilities in $PM_{2.5}$ sensitivities to 40% reductions in emissions of $NH_3$, $NO_x$, and $SO_x$ individually, and collectively, for the four world regions. The regionally averaged $PM_{2.5}$ sensitivities are summarised in Fig. 10 and Table S5. $PM_{2.5}$ concentrations in East Asia show comparable sensitivities to individual emissions reductions in $NH_3$, $NO_x$, and $SO_x$, with the impacts of $NH_3$ and $NO_x$ emissions reductions

being more concentrated in continental areas than those for $SO_x$ emissions reductions (Fig. 9). A 40% reduction in $NO_x$ emissions yields a maximum decrease in $PM_{2.5}$ of 11.5 $\mu g\ m^{-3}$ (12%) over southern China, while 40% reductions in $NH_3$ and $SO_x$ yield slightly smaller maximum $PM_{2.5}$ decreases of 8.51 $\mu g\ m^{-3}$ (10%) and 8.78 $\mu g\ m^{-3}$ (9%), respectively. The regional average sensitivities of $PM_{2.5}$ concentrations in East Asia to 40% reductions in individual precursors are ~8% (Fig. 10).

In contrast, $PM_{2.5}$ concentrations in South Asia only show significant responses to $SO_x$ emissions reductions, whilst $NH_3$ emissions reductions have little effect, which is consistent with our previous finding that South Asia has the most ammonia-rich chemical climate for SIA formation (Ge et al., 2022). The dominant proportion of $(NH_4)_2SO_4$ in SIA compared to $NH_4NO_3$ in South Asia (Fig. S8) also explains the small sensitivity of $PM_{2.5}$ in this region to $NO_x$ emissions reductions. The maximum $PM_{2.5}$ decrease (7.02 $\mu g\ m^{-3}$, 9%) for 40% $SO_x$ emissions reductions is more than three times larger than the maximum $PM_{2.5}$ decrease (2.20 $\mu g\ m^{-3}$, 4%) for 40% $NH_3$ emissions reductions. The decreases in regionally averaged $PM_{2.5}$ concentrations in South Asia in response to 40% reductions in emissions of individual precursors are in the order 10% for $SO_x$, 3% for $NO_x$, and 1% for $NH_3$.

In the Euro_Medi region, $PM_{2.5}$ sensitivities vary from north to south. Northern and central Europe is most sensitive to $NH_3$ and $NO_x$ emissions reductions, for which the maximum decrease in $PM_{2.5}$ is ~2.6 $\mu g\ m^{-3}$ (16%) for 40% reductions, while the Mediterranean is more sensitive to $SO_x$ emissions reductions, for which the maximum decrease in $PM_{2.5}$ is 2.98 $\mu g\ m^{-3}$ (12%) for 40% reductions. Regionally averaged, however, the $PM_{2.5}$ concentrations in Euro_Medi show comparable sensitivities to the three precursors with decreases in the range 5-8% for 40% emissions reductions in individual precursors.

Over North America, the eastern US shows larger sensitivities of $PM_{2.5}$ to all emissions reduction scenarios than the western US, and reductions in $NO_x$ emissions yield larger decreases in $PM_{2.5}$ than reductions in $SO_x$ and $NH_3$ emissions. The maximum decrease in $PM_{2.5}$ derived from 40% reductions in $NO_x$ emissions is 3.10 $\mu g\ m^{-3}$ (16%); for 40% reductions in $NH_3$ and $SO_x$ emissions the maximum $PM_{2.5}$ decreases are 2.37 $\mu g\ m^{-3}$ (14%) and 1.35 $\mu g\ m^{-3}$ (10%) respectively. The regional average sensitivities of $PM_{2.5}$ concentrations in North America to 40% reductions in emissions of individual precursors decreases in a slightly different order: $NO_x$ (8%), $SO_x$ (7%), and $NH_3$ (4%).

Figure 10 shows that 20% emissions reductions in any precursor lead to decreases in regionally averaged $PM_{2.5}$ concentrations, although the $PM_{2.5}$ sensitivities vary substantially with precursor and region. Given the non-one-to-one chemical responses of SIA components to reductions in emissions in individual precursors discussed in Sect. 3.1, even 20% reductions appear substantial enough to ensure that decreased SIA formation due to decreased precursor emissions dominates over any disbenefits to SIA formation from, for example, increases in oxidant levels induced by $NO_x$ emissions reductions. For instance, 20% reductions in $NO_x$ emissions still cause a decrease of 0.77 $\mu g\ m^{-3}$ (3%) in regional average $PM_{2.5}$ in East Asia, despite increasing regional average $SO_4^{2-}$ by 0.10 $\mu g\ m^{-3}$ (2%) because it decreases regional average $NH_4^+$ and fine $NO_3^-$ by greater amounts (0.17 $\mu g\ m^{-3}$ (5%) and 0.72 $\mu g\ m^{-3}$ (14%) respectively). Similarly, 20% reductions in $SO_x$ emissions decrease regional average $PM_{2.5}$ in East Asia by 1.15 $\mu g\ m^{-3}$ (4%) because the decreases in $SO_4^{2-}$ (1.06 $\mu g\ m^{-3}$, 19%) and $NH_4^+$ (0.32 $\mu g\ m^{-3}$, 9%) caused by reduced $(NH_4)_2SO_4$ formation are larger than the increase in fine $NO_3^-$ (0.20 $\mu g\ m^{-3}$, 4%) due to elevated $NH_4NO_3$ formation. On the other hand, the mitigation of $PM_{2.5}$ by reducing emissions of all 3 precursors together is impacted by these non-one-to-one chemical responses as well, which causes the net decrease in regional average $PM_{2.5}$ derived from reductions in all 3 precursors to be smaller than the sum of individual $PM_{2.5}$ decreases derived from reductions in emissions of precursors individually. For example, 40% reductions in $NH_3$, $NO_x$ and $SO_x$ emissions individually decrease regional average $PM_{2.5}$ in East Asia by 2.03 $\mu g\ m^{-3}$ (7%), 1.89 $\mu g\ m^{-3}$ (7%), and 2.33 $\mu g\ m^{-3}$ (8%) respectively (sum of the three: 6.25 $\mu g\ m^{-3}$), while the decrease in regional average $PM_{2.5}$ derived from 40% reduction in all 3 precursors simultaneously is 5.59 $\mu g\ m^{-3}$ (20%).

The 40% reduction in emissions of all 3 species yields a maximum decrease in $PM_{2.5}$ of 23.9 $\mu g\ m^{-3}$ (25%) over East Asia, followed by 10.4 $\mu g\ m^{-3}$ (17%) in South Asia, 5.57 $\mu g\ m^{-3}$ (22%) in Euro_Medi, and 5.05 $\mu g\ m^{-3}$ (28%) in North America.

The regional average sensitivity of $PM_{2.5}$ concentrations to 20% and 40% reductions in emissions of all 3 species decreases in the order East Asia (10% and 20% for 20% and 40% reductions respectively), Euro_Medi (9% and 17%), North America (8% and 17%), and South Asia (7% and 13%). This trend is related to differences in the contribution of SIA to $PM_{2.5}$ in the different regions. Figure S8 shows the mass contributions of individual $PM_{2.5}$ components to the regional average concentration of $PM_{2.5}$ in the baseline and the 40% emissions reductions scenarios. SIA components in the baseline account for over half of $PM_{2.5}$ in East Asia (52%), followed by Euro_Medi (42%), North America (35%), and South Asia (31%), which explains why reductions in emissions of all three SIA precursors are most efficient for the mitigation of $PM_{2.5}$ in East Asia but least efficient in South Asia. In fact, primary $PM_{2.5}$ is the largest contributor to $PM_{2.5}$ in South Asia, so reducing these emissions will be the most efficient way of abating $PM_{2.5}$ pollution in this region. It is noteworthy that in the scenario of 40% reductions in all 3 species, SIA is still the largest contributor to $PM_{2.5}$ in East Asia, while primary $PM_{2.5}$ and Rest (mainly secondary organic aerosol) become the dominant contributors in other regions. Even with 40% reductions in all three SIA precursors, none of the four regions has its regional average $PM_{2.5}$ concentration decreased to below 5 µg m$^{-3}$. Euro_Medi (8.4 µg m$^{-3}$ after 40% reductions) and North America (6.5 µg m$^{-3}$) are the closest, whilst East Asia (21.8 µg m$^{-3}$) and South Asia (27.0 µg m$^{-3}$) are still far away from achieving the latest WHO AQG for $PM_{2.5}$. Therefore, reductions in emissions of primary $PM_{2.5}$ and in VOCs are also required to achieve further $PM_{2.5}$ reductions in all regions, or even greater reductions in SIA precursors than simulated here.

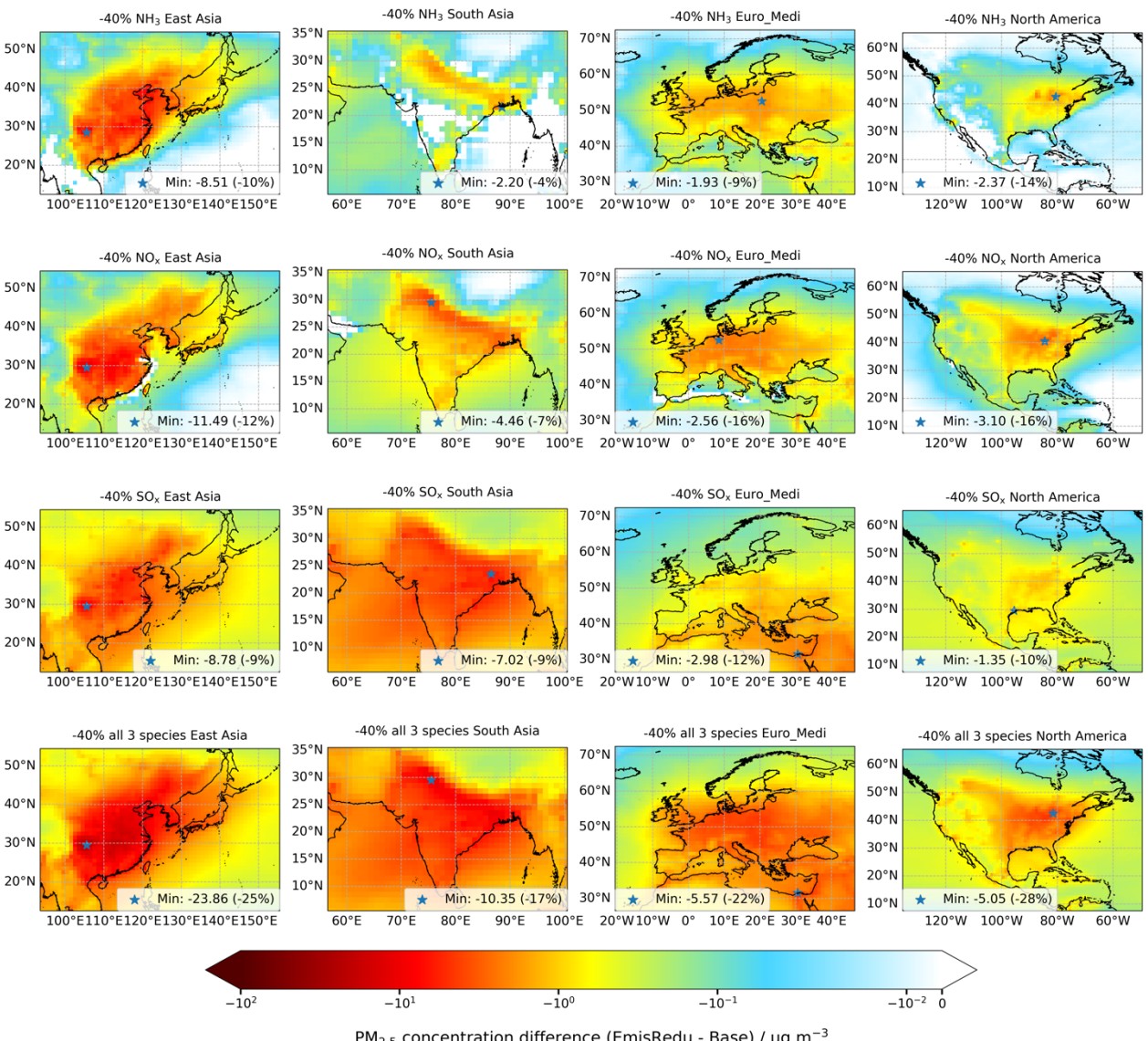

**Figure 9: Changes in $PM_{2.5}$ annual mean surface concentrations for 40% emissions reductions in NH₃, NOₓ, and SOₓ individually and collectively. The blue star in each map locates the minimum difference within each region.**

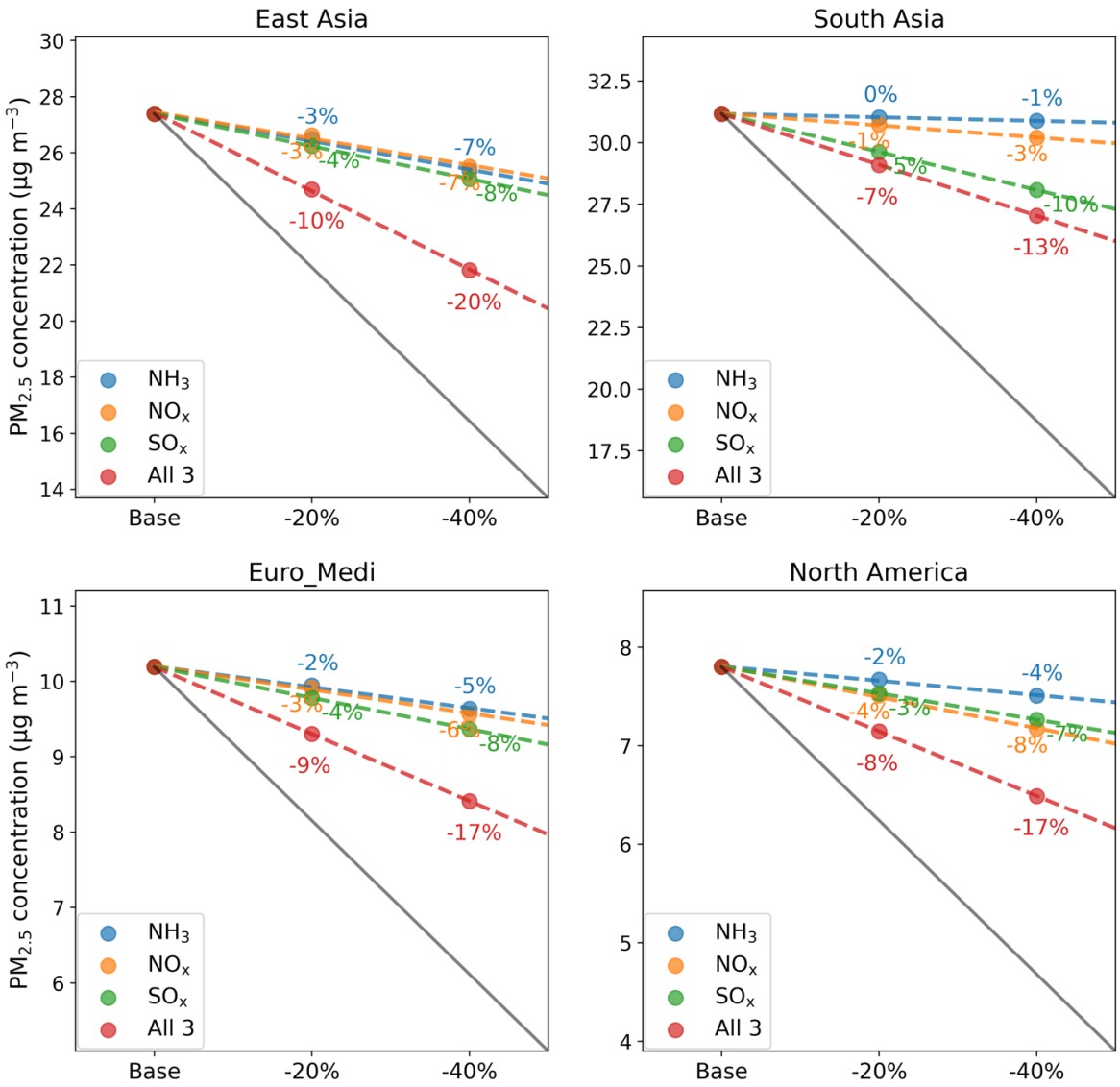

**Figure 10: The absolute and relative sensitivities of regionally-averaged annual mean surface concentrations of PM$_{2.5}$ to 20% and 40% emissions reductions in NH$_3$ (blue), NO$_x$ (orange) and SO$_x$ (green) individually, and collectively (red), for the four regions defined in Fig. 1. The solid grey line in each panel illustrates the one-to-one relative response to emissions reductions, whilst the coloured dashed lines are the linear regressions through each set of three model simulations and illustrate the actual responses to emissions reductions of a given precursor. The numbers show the corresponding relative responses to each emissions reduction (with respect to baseline).**

### 3.3 Sensitivity of N$_r$ and S$_r$ deposition

The impacts of reductions in emissions of NH$_3$, NO$_x$ and SO$_x$ on total amounts of N and S deposition are straightforward because these must match the emissions mass changes in N and S. However, the relative amounts and spatial pattern of the individual components of N and S deposition are impacted.

Figure 11 shows the spatial variations in the sensitivities of the wet and dry deposition of the RDN components NH$_3$ and NH$_4^+$ to 40% reductions in global emissions of NH$_3$, NO$_x$, SO$_x$ individually, and collectively. Figure 12 shows similar for deposition of the OXN components NO$_x$, HNO$_3$, TNO$_3^-$ (total NO$_3^-$, the sum of fine and coarse NO$_3^-$) and Rest (the sum of other oxidized N species). Both wet and dry deposition of NH$_3$ show negative responses to emissions reductions in NH$_3$ and all 3 precursors together, but positive responses to emissions reductions in NO$_x$ and SO$_x$, which is consistent with the responses of surface NH$_3$ concentrations to these emissions reductions (Sect. 3.1.1). This is because reduced NO$_x$ and SO$_x$ emissions lead to decreased concentrations of acidic species in the atmosphere, resulting in more NH$_3$ remaining in the gas phase and

485 greater $NH_3$ deposition over continents. In contrast, global $NH_4^+$ wet and dry deposition decreases in all emissions reduction scenarios, which is also in line with the decreased $NH_4^+$ concentrations in all scenarios.

The changes in deposition of the OXN components to emissions reductions are more complicated (Fig. 12). The two species with the largest variation in deposition across the emissions reduction scenarios are $HNO_3$ and $TNO_3^-$, which is due to their large contributions to total OXN deposition in most world regions (Ge et al., 2022). For reductions in emissions of $NO_x$
and all 3 species, all OXN deposition components show clear decreasing trends due to the strong reduction in their precursor emissions.

In response to $NH_3$ emissions reductions, Fig. 12 shows that $HNO_3$ wet and dry deposition increases in eastern China, northern India, Europe, and eastern North America, whereas the wet and dry deposition of $TNO_3^-$ decreases in these regions. Further examination of fine and coarse $NO_3^-$ deposition differences in Fig. S9 shows that the decrease in $TNO_3^-$ deposition is
495 driven by the decrease in fine $NO_3^-$ wet and dry deposition, while coarse $NO_3^-$ wet and dry deposition in the four regions actually increases. The reduction in $NH_3$ emissions decreases $NH_4NO_3$ formation and therefore liberates more $HNO_3$; as a result, more OXN deposits in the form of $HNO_3$ rather than $NO_3^-$ in these regions. In contrast, impacts of $SO_x$ emissions reductions on $HNO_3$ and $TNO_3^-$ deposition are the opposite of impacts of $NH_3$ emissions reductions. As discussed in Sect. 3.1.2, decreased $SO_4^{2-}$ concentrations promote the formation of $NH_4NO_3$, which results in decreased wet and dry deposition
of $HNO_3$ and consequently increased $TNO_3^-$ deposition (driven by increased fine $NO_3^-$ deposition, Fig. S9) in eastern China, northern India, Europe, and eastern North America.

Compared to $HNO_3$ and $TNO_3^-$, the responses of $NO_x$ and Rest OXN deposition to reductions in $NH_3$ and $SO_x$ emissions are considerably smaller. For instance, the maximum increases in $HNO_3$ wet and dry deposition in response to 40% $NH_3$ emissions reductions are 328 mgN m$^{-2}$ (85%) and 248 mgN m$^{-2}$ (48%) respectively, whereas the maximum increases in Rest
OXN wet and dry deposition are only 1.35 mgN m$^{-2}$ (4%) and 1.64 mgN m$^{-2}$ (4%) respectively. Also, in the 40% $NH_3$ emissions reduction scenario, the maximum decreases in $TNO_3^-$ wet and dry deposition are 359 mgN m$^{-2}$ (25%) and 175 mgN m$^{-2}$ (28%) respectively, whereas the maximum decreases in $NO_x$ dry deposition are only 4.22 mgN m$^{-2}$ (1%).

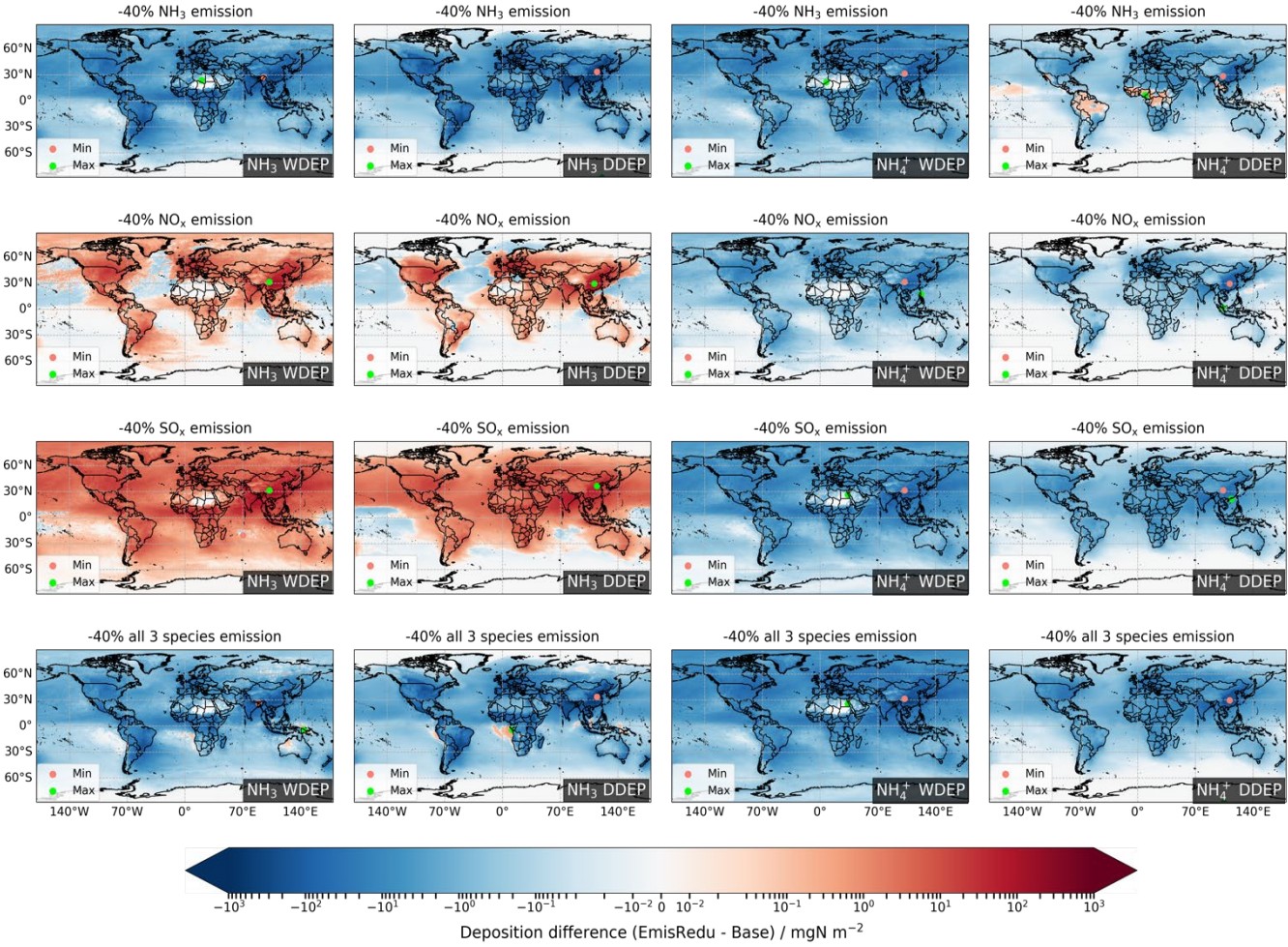

**Figure 11: Changes in wet (WDEP) and dry deposition (DDEP) of NH₃ and NH₄⁺ for 40% emissions reductions in NH₃, NOₓ, and SOₓ individually and collectively. Red and green dots in each map locate the minimum and maximum difference, respectively.**

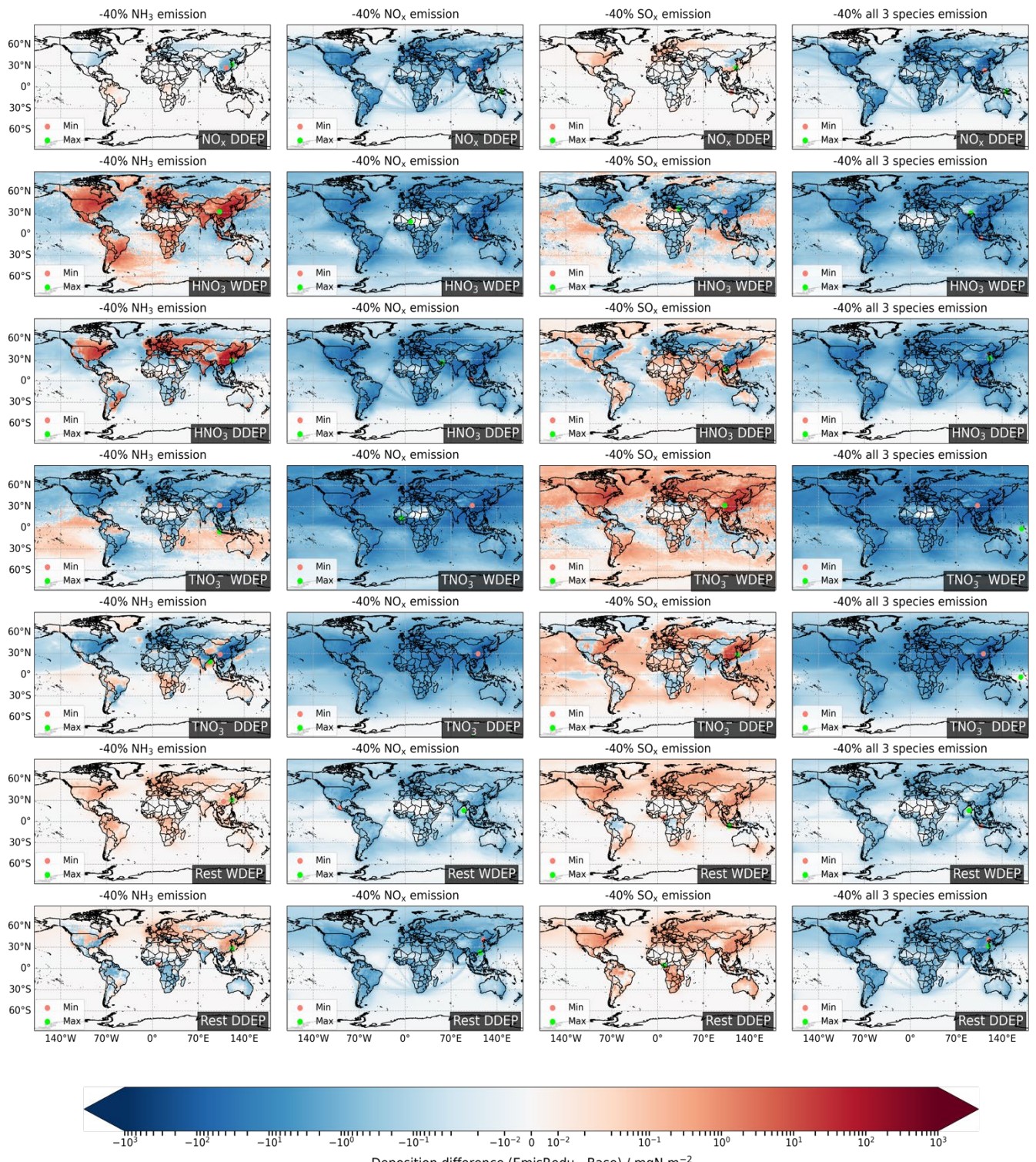

Figure 12: Changes in wet (WDEP) and dry deposition (DDEP) of NO$_x$, HNO$_3$, TNO$_3^-$ (fine + coarse NO$_3^-$), and Rest OXN species for 40% emissions reductions in NH$_3$, NO$_x$, and SO$_x$ individually and collectively. Red and green dots in each map locate the minimum and maximum difference, respectively.

The spatial variations in the sensitivities of SO$_2$ and SO$_4^{2-}$ wet and dry deposition to the precursor emissions reductions are shown in Fig. 13. Increased SO$_2$ wet deposition is observed globally in response to NH$_3$ emissions reductions, whilst SO$_2$ dry deposition decreases over the continents but increases over the oceans. The non-stomatal canopy resistance of SO$_2$ is positively correlated to the molar acidity ratio $a_{SN}$ (Smith et al., 2000; Erisman et al., 2001; Fowler et al., 2009; Massad et al., 2010; Simpson et al., 2012). Reductions in NH$_3$ emissions increase the $a_{SN}$, which therefore increases the canopy resistance

525 of SO$_2$ and decreases SO$_2$ dry deposition over the continents. The 40% reduction in NH$_3$ emissions yields a global maximum increase in SO$_2$ wet deposition of 131 mgS m$^{-2}$ (44%), and a maximum decrease in SO$_2$ dry deposition of 600 mgS m$^{-2}$ (24%). The overall effect of increased SO$_2$ wet deposition and decreased SO$_2$ dry deposition is a decreased SO$_2$ total deposition over populated continents (where NH$_3$ emissions are high) and increased SO$_2$ total deposition over oceans (Fig. S5). The maximum decrease of SO$_2$ total deposition located in southern China is 586 mgS m$^{-2}$ (18%) for 40% NH$_3$ reduction.

530 The sensitivity of total deposition of SO$_4^{2-}$ to NH$_3$ emissions reductions (Fig. S6) follows the trend in the sensitivity of SO$_4^{2-}$ concentrations (Sect. 3.1.3). The responses of wet and dry deposition of SO$_4^{2-}$ to NH$_3$ emissions reductions are similar (Fig. 13). In general, decreased wet and dry SO$_4^{2-}$ deposition appears over Europe, north-eastern China, and north-eastern US, while increased wet and dry deposition occurs in the rest of the world. As also for their concentration sensitivities, the magnitudes of the SO$_4^{2-}$ deposition responses are much smaller than for SO$_2$. For 40% NH$_3$ emissions reduction, the maximum decrease of SO$_4^{2-}$ total deposition is 42 mgS m$^{-2}$ (9%), which is an order of magnitude smaller than that of SO$_2$ total deposition.

For NO$_x$ emissions reductions, both wet and dry deposition of SO$_2$ generally show decreases in eastern China, Europe, and north-eastern US but increases in the rest of the world, which is contrary to the responses of SO$_4^{2-}$ wet and dry deposition. This is related to enhanced chemical conversion of SO$_2$ to SO$_4^{2-}$ due to increased atmospheric oxidizing capacity over eastern China, Europe, and north-eastern US (details in Sect. 3.1.3). The maximum decrease in SO$_2$ wet and dry deposition in response 540 to 40% reductions in NO$_x$ emissions is ~65 mgS m$^{-2}$, while the maximum increase in SO$_4^{2-}$ wet and dry deposition is ~50 mgS m$^{-2}$.

For reductions in SO$_x$ and in all 3 precursors collectively, decreased wet and dry deposition of SO$_2$ and SO$_4^{2-}$ is observed globally and the 40% reductions in these two scenarios yield similar global maximum decreases in SO$_2$ deposition (wet: ~937 mgS m$^{-2}$, 41%; dry: ~1338 mgS m$^{-2}$, 43% ) and SO$_4^{2-}$ deposition (wet: ~828 mgS m$^{-2}$, 38%; dry: ~150 mgS m$^{-2}$, 39%).

545

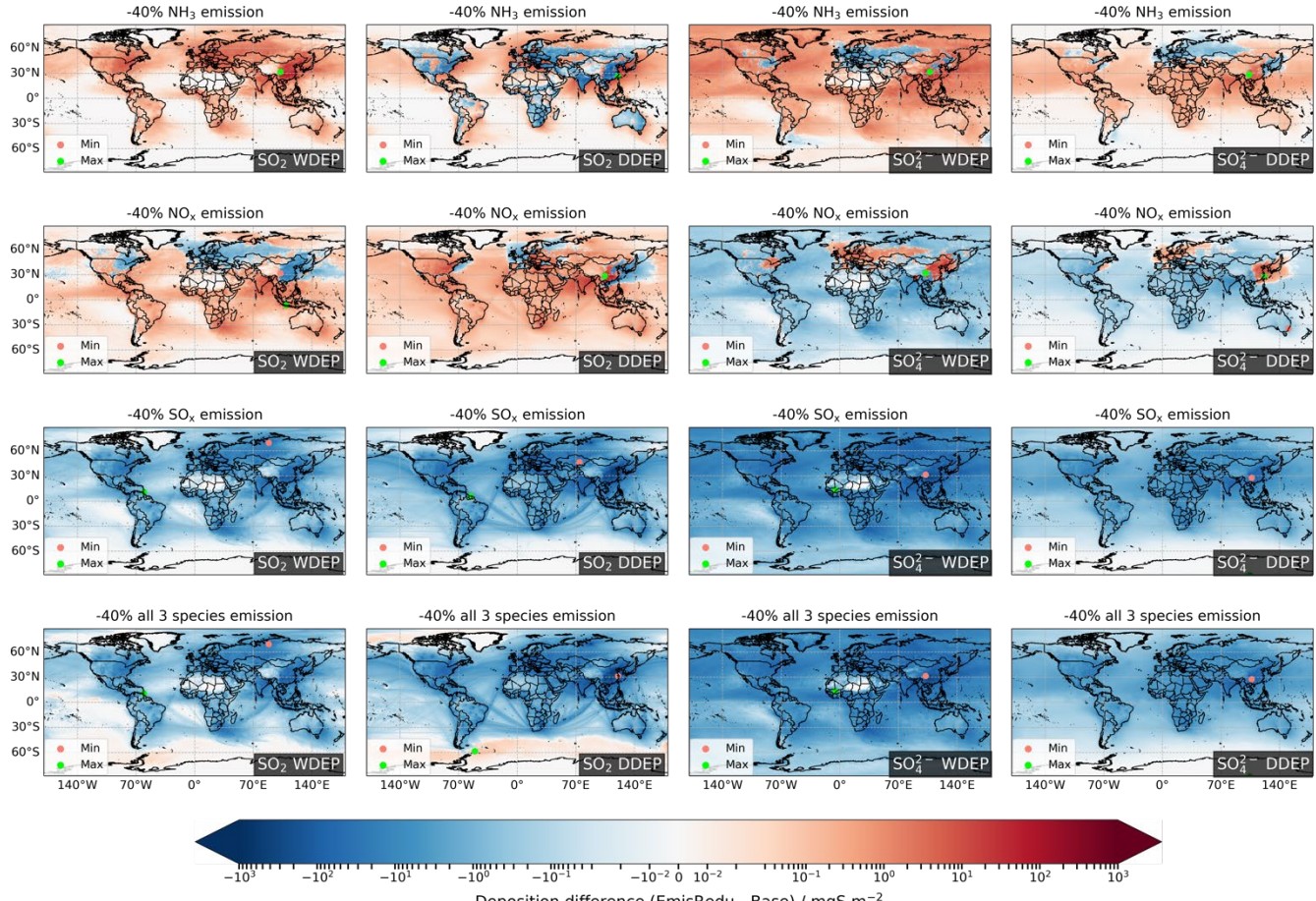

**Figure 13: Changes in wet (WDEP) and dry deposition (DDEP) of SO$_2$ and SO$_4^{2-}$ for 40% emissions reductions in NH$_3$, NO$_x$, and SO$_x$ individually and collectively. Red and green dots in each map locate the minimum and maximum difference, respectively.**

Figure 14 shows the differences of regional total deposition of individual species between baseline and the 40% emissions reduction scenarios. The regional total RDN, OXN and OXS deposition and the contributions of individual components are listed in Table S6 and S7. As expected, the responses of regional total RDN, OXN and OXS deposition are essentially linear through 20% and 40% reductions in their corresponding precursor, and largely insensitive to reductions in the other precursors, although there are slight differences between regions. For example, 20% and 40% reductions in $NH_3$ emissions respectively yield 20-21% and 39-41% decreases in total deposition of RDN in the four regions, whilst having no effect on regional total OXN and OXS deposition. The contributions of different deposition components do, however, vary with emission changes and the different lifetimes of the deposition components contribute to the small variabilities in responses of total RDN, OXN and OXS deposition to emissions reductions via differences in the transport through the regional boundaries.

A 40% reduction in $NH_3$ emissions produces a decrease of 3.58 TgN yr$^{-1}$ (51% of regional total deposition of the same species in baseline; similarly, hereinafter) in $NH_3$ total deposition and 1.16 TgN yr$^{-1}$ (24%) in $NH_4^+$ total deposition over East Asia (Fig. 14), which causes the contribution of $NH_3$ to RDN deposition to decrease from 59% (baseline) to 47% (after 40% $NH_3$ reduction). In other regions, reductions in $NH_3$ emissions also decrease $NH_3$ deposition more effectively than $NH_4^+$ deposition. Also in East Asia, a 40% $NH_3$ emissions reduction increases $HNO_3$ deposition by 0.61 TgN yr$^{-1}$ (20%) and $SO_4^{2-}$ deposition by 0.06 TgS yr$^{-1}$ (2%) but decreases $TNO_3^-$ deposition by 0.55 TgN yr$^{-1}$ (14%) and $SO_2$ deposition by 0.12 TgS yr$^{-1}$ (3%). As a result, the contribution of $HNO_3$ deposition to total OXN deposition increases from 30% to 37%, corresponding to a decrease in $TNO_3^-$ contribution from 54% to 47%, whereas changes in contributions of $SO_2$ and $SO_4^{2-}$ to OXS deposition are very small (Table S7). In other world regions, such changes in total deposition of $HNO_3$, $TNO_3^-$, $SO_2$, and $SO_4^{2-}$ derived from $NH_3$ emissions reductions are similar but of smaller magnitude.

For 40% $NO_x$ emissions reductions, $TNO_3^-$ deposition shows the largest decrease in East Asia (1.57 TgN yr$^{-1}$, 35%), South Asia (0.44 TgN yr$^{-1}$, 26%), and Euro_Medi (0.71 TgN yr$^{-1}$, 37%), whilst $HNO_3$ deposition shows the largest decrease in North America (0.56 TgN yr$^{-1}$, 40%). In contrast, the sensitivities of $NO_x$ dry deposition to $NO_x$ emissions reductions are very small. The contributions of individual OXN deposition components remain fairly constant in all regions. Furthermore, East Asia, Euro_Medi, and North America show a 5% increase in the contribution of $NH_3$ deposition to total RDN deposition and a corresponding 5% decrease in $NH_4^+$ contribution for 40% $NO_x$ emissions reductions, which reflects a small shifting of gas-aerosol partitioning for RDN as well. This kind of contribution change in RDN deposition is 2% for South Asia. The impacts of $NO_x$ emissions reductions on OXS deposition compositions are very small.

The 40% reductions in $SO_x$ emissions yield 3.24 TgS yr$^{-1}$ (39%), 1.07 TgS yr$^{-1}$ (38%), 1.17 TgS yr$^{-1}$ (33%), and 0.78 (37%) TgS yr$^{-1}$ decreases in OXS deposition over East Asia, South Asia, Euro_Medi, and North America respectively. $SO_x$ emissions reductions cause larger decreases in $SO_2$ deposition than in $SO_4^{2-}$ deposition in East Asia and Euro_Medi, while $SO_2$ and $SO_4^{2-}$ deposition in South Asia and North America show similar sensitivities. This is associated with slightly greater proportions of $SO_2$ (56-58%) to OXS deposition in the former regions than in the latter regions, and that these proportions are not affected by $SO_x$ emissions reductions (Table S7). $NH_3$ and $NH_4^+$ deposition is moderately sensitive to $SO_x$ emissions reductions in the four regions. An increase of 0.75 TgN yr$^{-1}$ (11%) in $NH_3$ total deposition and a decrease of 0.71 TgN yr$^{-1}$ (14%) in $NH_4^+$ total deposition for 40% reductions in $SO_x$ emissions is observed over East Asia. For South Asia, Euro_Medi, and North America, the increases in $NH_3$ total deposition due to 40% $SO_x$ reductions are 0.51 TgN yr$^{-1}$ (12%), 0.25 TgN yr$^{-1}$ (11%), and 0.23 TgN yr$^{-1}$ (12%) respectively. This is because reduced $SO_x$ emissions lead to reductions in $(NH_4)_2SO_4$ formation which then cause increased $NH_3$ but decreased $NH_4^+$ concentrations. Another side effect of $SO_x$ emissions reductions in East Asia is a slight decrease in $HNO_3$ deposition (0.25 TgN yr$^{-1}$, 10%) and an equivalent increase in $TNO_3^-$ deposition (0.21 TgN yr$^{-1}$, 5%). The equivalent deposition changes are considerably smaller in the other three regions, which again indicates a larger amount of $NH_4NO_3$ pollution in East Asia than other regions.

For collective reductions in emissions of all precursors, the changes in deposition of each species reflect net effects of individual reductions in emissions of $NH_3$, $NO_x$, and $SO_x$. For instance, Fig. 14 shows that the decrease in $NH_3$ deposition in

East Asia derived from 40% reduction in emissions of all 3 species (2.58 TgN yr$^{-1}$, 36%) is smaller than that from individual NH$_3$ emissions reduction (3.58 TgN yr$^{-1}$, 51%) due to the compensating effects of simultaneous NO$_x$ and SO$_x$ emissions reductions in the former scenario. In contrast, the decrease in NH$_4^+$ deposition in East Asia for 40% emissions reductions in all 3 species (2.11 TgN yr$^{-1}$, 43%) is almost double that from individual NH$_3$ emissions reduction scenario (1.16 TgN yr$^{-1}$, 24%). The variations in chemical forms of RDN, OXN, and OXS deposition affect where they deposit as well since N$_r$ and S$_r$ species have different lifetimes and a shorter lifetime causes a more localised deposition. Many studies show that NH$_3$ and HNO$_3$ have shorter lifetimes than NH$_4^+$ and NO$_3^-$ (Xu and Penner, 2012; Hauglustaine et al., 2014; Bian et al., 2017; Ge et al., 2022). The abatement of total N (TN = RDN + OXN) and S deposition within a certain region is partially offset by this more localised deposition pattern especially in South Asia and Euro_Medi. The 40% reductions in all 3 species emissions yield a 34% (2.93 TgN yr$^{-1}$) decrease in regional TN deposition in South Asia, and a 34% (1.18 TgS yr$^{-1}$) decrease in regional OXS deposition in Euro_Medi, which means that less N$_r$ and S$_r$ pollution is transported outside these regions, and more is deposited locally.

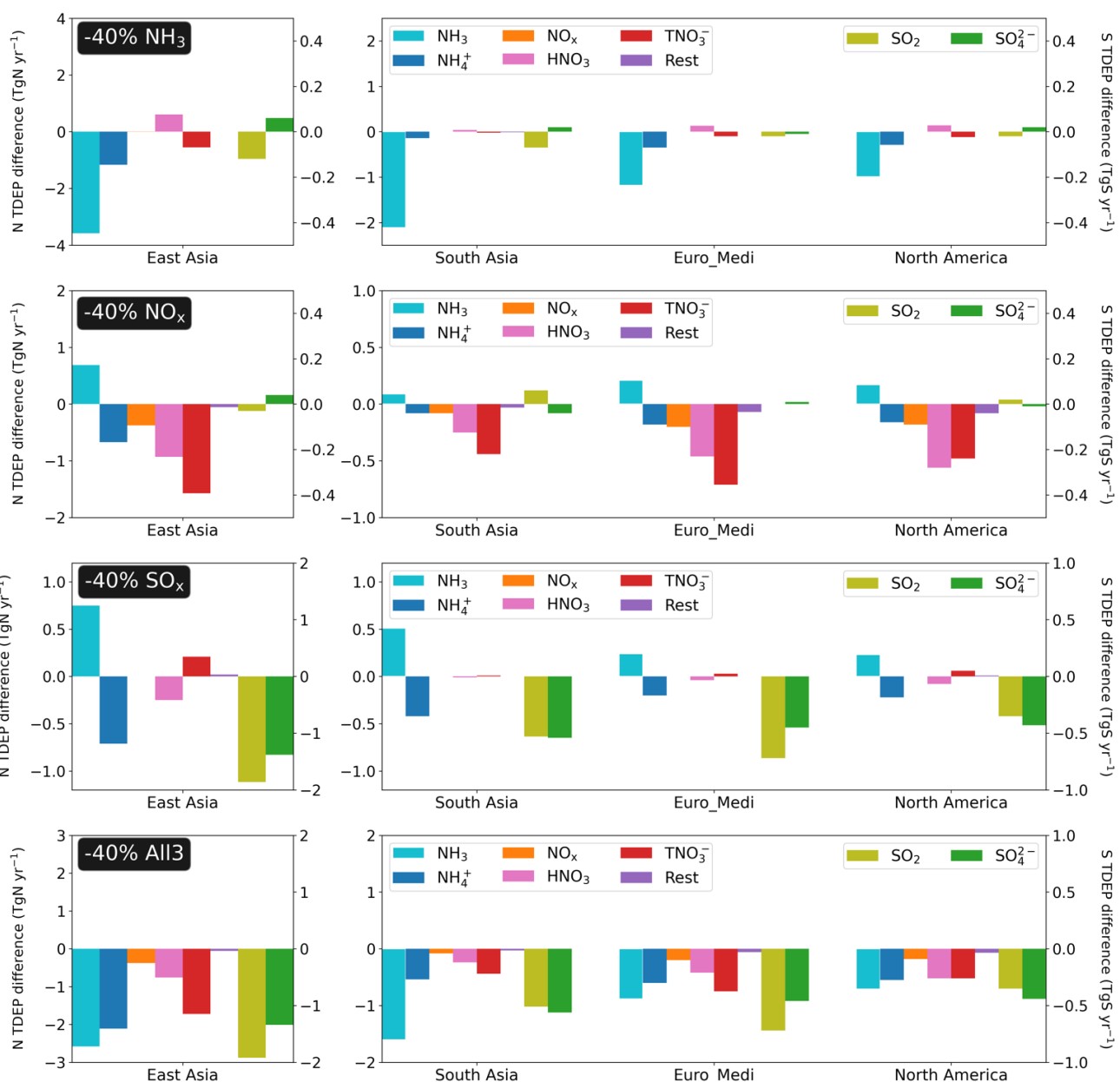

**Figure 14: The absolute sensitivities (Emission Reduction - Baseline) of regional total deposition (wet + dry) of NH$_3$, NH$_4^+$, NO$_x$, HNO$_3$, TNO$_3^-$ (fine + coarse NO$_3^-$), SO$_2$ and SO$_4^{2-}$ to 40% emissions reductions in NH$_3$ (top row), NO$_x$ (2$^{nd}$ row) and SO$_x$ (3$^{rd}$ row) individually, and collectively (bottom row), for the four regions defined in Fig. 1. The left-hand $y$ axis in each panel is for RDN and OXN species, while the right-hand $y$ axis is for OXS species.**

## 4 Discussions

### 4.1 Model uncertainty

This study uses global model simulations for 8 emissions reduction scenarios to investigate the geographical variation in the sensitivities of $N_r$, $S_r$, and $PM_{2.5}$ to emissions reductions in inorganic precursor gases. Although the EMEP MSC-W ACTM is state-of-the-art and widely used in scientific research and policy development, the analyses presented in this study are based on data from a single modelling system, and their accuracy is therefore subject to any uncertainties in the EMEP MSC-W ACTM's specific parameterisation of atmospheric processes, as well as uncertainties in the chosen emission inventory and meteorology input. However, the evaluation of surface concentrations and wet deposition of $N_r$ and $S_r$ species from this model configuration for the same year against global measurements from 10 monitoring networks (Ge et al., 2021b) has demonstrated the model's capability to capture the overall spatial variations in annual concentrations of $NH_3$, $NH_4^+$, $NO_2$, $HNO_3$, fine $NO_3^-$, $SO_2$ and $SO_4^{2-}$ and their wet deposition in East Asia, Southeast Asia, Europe, and North America. No nationally compiled data for these species is publicly available in Oceania, South Asia, Africa, and Latin America, which inhibits model evaluation in these regions. Similarly, the lack of dry deposition comparison is due to the lack of measurements.

Ge et al. (2021b) also evaluate model outputs derived from two global emission inventories and finds large localised discrepancies in $NH_3$, $NO_2$ and $SO_2$ emissions in certain world regions between the two inventories. The impacts of these discrepancies on modelled concentrations varies with regard to primary or secondary species and differs regionally. Meanwhile, Ge et al. (2021b) indicate that the model shows generally better linear correlations with measurement networks in Southeast Asia ($\bar{R} = 0.73$ over 7 species), Europe ($\bar{R} = 0.67$ over 7 species) and North America ($\bar{R} = 0.63$ over 7 species) than in China ($\bar{R} = 0.35$ over 5 species), which implies potential discrepancies among different measurements or regional emissions rather than systematic issues with the model parameterisation. The uncertainties in both model and measurement constrains the extent to which the agreement between model and measurement can be used to evaluate a model's performance. The values of statistical metrics for this EMEP-WRF system are at least as good as other global modelling studies (Bian et al., 2017; Hauglustaine et al., 2014). As already indicated, a detailed discussion of model and measurement uncertainties is presented in Ge et al. (2021b).

Additionally, the sensitivities of global and regional $N_r$, $S_r$, and $PM_{2.5}$ to various emissions reductions can only be investigated through modelling experiments, and since most model uncertainties will be similar across a set of simulations with the same model configuration, the modelled changes in concentrations between baseline and an emissions reduction scenario should be robust. Nevertheless, considering the fundamental uncertainties in emissions and model parameterizations, all numbers reported in this work should be considered as having underlying uncertainty, albeit that the latest available global emissions inventory and model version were used to minimize the impacts of these uncertainties.

### 4.2 Regional co-benefits and disbenefits of single-precursor emissions controls

The Results section shows that the reduction in emissions of one individual precursor has multiple co-benefits and sometimes small disbenefits on mitigating $N_r$, $S_r$, and $PM_{2.5}$ pollution, and these effects are geographically variable. In this work, our discussion focuses on East Asia, South Asia, Euro_Medi, and North America on account of both the thoroughly evaluated model results and the high population density plus high $N_r$ and $S_r$ pollution in these regions. The comparison of regional responses to emissions reductions reveals differences in regional oxidation regime, SIA chemistry and deposition pattern which are important processes to consider when designing emissions control policies since transitory increases in $PM_{2.5}$ and some $N_r$ and $S_r$ pollutants could occur as emissions reduction measures are gradually applied.

Globally, reductions in $NH_3$ emissions are effective for reducing $NH_3$ concentrations and its wet and dry deposition but considerably less effective at reducing $NH_4^+$. This is because most world regions are in an ammonia-rich chemical domain in which reducing $NH_3$ emissions only has limited effects on mitigating SIA formation (Ge et al. 2022). Other co-benefits of $NH_3$

emissions reductions include reductions in fine $NO_3^-$ surface concentrations and deposition in East Asia, South Asia, Euro_Medi, and North America because of reduced $NH_4NO_3$ formation. A notable disbenefit is the increased $SO_2$ surface concentration and human exposure in these regions which is caused by reduced $SO_2$ dry deposition. The dry deposition velocity of $SO_2$ is negatively correlated with the molar acidity ratio $a_{SN}$ which is a model parameterisation derived from long-term deposition measurements (Erisman et al., 2001; Simpson et al., 2012). Reduced $NH_3$ emissions therefore increase the acidity, leading to decreased $SO_2$ dry deposition.

Similarly, whilst reducing $NO_x$ emissions is of course an effective way of decreasing global concentrations and deposition of OXN species, the degree to which different OXN species are decreased varies across regions. A 40% reduction in $NO_x$ emissions decreases $NO_x$ and fine $NO_3^-$ surface concentrations in East Asia by 45% and 33% respectively, whereas in South Asia this measure has a greater effect on fine $NO_3^-$ (45% decrease) than on $NO_x$ (22% decrease). In Euro_Medi and North America, the 40% $NO_x$ emissions reductions produce similar decreases in regional average $NO_x$ (36-38%) and fine $NO_3^-$ (41-42%) concentrations. These varying regional responces are consequent on different regional $NO_x$ oxidation regime and SIA chemistry. The $NO_x$ emissions reductions decrease $NO_x$ surface concentrations, which increases $O_3$ concentrations in the high $NO_x$ areas of eastern China and western and central Europe and therefore increases the atmospheric oxidizing capacity in these regions. As a result, more $SO_2$ is oxidized to $SO_4^{2-}$ which leads to decreased $SO_2$ concentrations and deposition and consequently increased $SO_4^{2-}$ concentrations and deposition in these areas. The enhanced $SO_4^{2-}$ production can partially (or even totally) offset the mass reduction in $PM_{2.5}$ caused by reduced $NH_4NO_3$ formation when reductions in $NO_x$ emissions are not sufficiently high. The increased oxidant levels will also enhance $HNO_3$ and $NO_3^-$ production in these regions, but this effect does not compensate for the reduction in $HNO_3$ and $NO_3^-$ concentrations due to the reductions in their precursors (at least for 20% and 40% $NO_x$ reductions), so the net effect is globally decreased $HNO_3$ and $NO_3^-$. Consequently, reduced $HNO_3$ and $NO_3^-$ levels caused by $NO_x$ emissions reductions lead to less $NH_4NO_3$ formation, which then results in globally increased $NH_3$ concentration and deposition and decreased $NH_4^+$ concentration and deposition. In contrast, decreased $O_3$ concentrations in South Asia and North America in response to $NO_x$ emissions reductions result in less chemical conversion of $SO_2$ to $SO_4^{2-}$, which then causes increased $SO_2$ and decreased $SO_4^{2-}$ concentrations and deposition. Clappier et al. (2021) and Thunis et al. (2021) showed that the increased atmospheric oxidizing capacity induced by reductions in $NO_x$ emissions is the reason for increased $PM_{2.5}$ levels in the Po basin (Italy) especially during wintertime, with increased nitrate, sulfate and SOA concentrations all being closely related to increased $O_3$ levels. Balamurugan et al. (2022) reported that reductions in SIA were much smaller than $NO_2$ emissions reductions during COVID lockdown in Germany, which is because the increased oxidant levels (OH, $NO_3$ and $O_3$) enhanced the formation of sulfate and night-time nitrate which then partially offset the lockdown-induced $PM_{2.5}$ decreases. Fu et al. (2020) noted that the increased oxidation of $NO_x$ to $HNO_3$ due to increased $O_3$ levels makes winter haze $NO_3^-$ in the North China Plain (NCP) almost insensitive to 30% reductions in emissions of $NO_x$, while Le et al. (2020) also revealed an unexpected PM exacerbation caused by unfavourable meteorological conditions and intensified SIA formation due to elevated $O_3$ levels induced by $NO_x$ emissions reductions during COVID lockdown in China.

The greatest effect of $SO_x$ emissions reductions is the direct decrease in global concentrations and deposition of $SO_2$ and $SO_4^2$, which then induces changes in gas-aerosol partitioning of $NH_3$-$NH_4^+$ and $HNO_3$-$NO_3^-$. As discussed above, the reduction in $(NH_4)_2SO_4$ formation frees more gaseous $NH_3$ and promotes $NH_4NO_3$ formation, leading to increased concentrations and deposition of $NH_3$ and fine $NO_3^-$ in all world regions. Considering that one $SO_4^{2-}$ takes up two $NH_3$ molecules under ammonia-rich conditions, but $NO_3^-$ only takes one, the net effect of $SO_x$ emissions reductions still causes globally decreased $NH_4^+$ concentrations and deposition. Liu et al. (2018) noted a significant increase in annual $NH_3$ concentrations caused by rapid $SO_2$ emissions reductions in the NCP. In addition, the shifting of RDN from aerosol-phase $NH_4^+$ to gaseous $NH_3$ in response to reduced $SO_x$ emissions also means that RDN pollution becomes more localized because $NH_3$ has a much shorter lifetime (1.6 days) than $NH_4^+$ (8.9 days) as it deposits more quickly to land rather than being transported to other regions (Ge et al. 2022). Utilizing combined measurements and modelling, Leung et al. (2020) found that the reduction in wintertime $PM_{2.5}$ in the NCP

is buffered by enhanced $NH_4NO_3$ formation due to decreased $SO_4^{2-}$ concentration liberating free $NH_3$, and increased oxidant levels promoting $HNO_3$ production, despite $SO_2$ and $NO_x$ emissions reductions in China. However, it is important to note that $(NH_4)_2SO_4$ has greater molecular mass than $NH_4NO_3$ and hence has larger leverage on $PM_{2.5}$ mass concentration, which ensures that the reductions in $PM_{2.5}$ mass concentrations derived from reduced $(NH_4)_2SO_4$ are less readily offset by increases in $NH_4NO_3$ concentrations. This also means that relative changes in the $SO_4^{2-}$ component cause greater mass changes with respect to $PM_{2.5}$ air quality objectives (which must be expressed as mass concentration) than do the same relative changes in the $NO_3^-$ component. The relative changes in the SIA components of $PM_{2.5}$ expressed as molar concentrations would be different.

Additionally, it is noteworthy that changes in $SO_x$ emissions have some subtle effects on $NO_x$ concentrations as well. According to Fig. 4, small increases in $NO_x$ concentrations are observed in southern China, western Europe, and eastern US, while India and north-eastern China show decreased $NO_x$ levels in response to $SO_x$ emissions reductions. These impacts can be attributed to the role of $SO_4^{2-}$ in the hydrolysis of $N_2O_5$ (which can undergo photolysis and produce $NO_2$). Many measurement studies, both laboratory and based on ambient samples, have revealed varying rates with different dependencies for this reaction such as the relative humidity, available aerosol surface area, the size and composition of the aerosol particles, and the ratio of sulfate to organic matter (Bertram et al., 2009; Bertram and Thornton, 2009; Chang et al., 2011; Wagner et al. 2013; McDuffie et al. 2018). The parameterization used in the EMEP MSC-W model, based on the work of Riemer et al. (2003), incorporates $SO_4^{2-}$ concentrations (via the parameter of aerosol surface area and reaction probability) into the calculation of the hydrolysis rate of $N_2O_5$ (Simpson et al. 2012). Consequently, reductions in $SO_x$ emissions affect $SO_4^{2-}$ levels which in turn affect $N_2O_5$ and $NO_x$ levels. While the changes in $NO_x$ concentrations are small when viewed at a regional average level (0-1%, as depicted in Figure 5), they may be significant in very local areas in the regions mentioned above. This part of the chemistry is one of the most uncertain aspects of the atmospheric science field. As a result, more laboratory and ambient measurement studies are needed to improve and test this type of model parametrization.

## 4.3 $PM_{2.5}$ sensitivities

This study highlights important messages for policy-makers with respect to regional $PM_{2.5}$ mitigation. The most effective emissions control for decreasing regional average $PM_{2.5}$ concentrations, via an individual component, differs between world regions, as follows.

More emissions controls focusing on $NH_3$ and $NO_x$ are necessary for regions with better air quality such as Europe and North America to decrease the annual $PM_{2.5}$ concentrations below the latest WHO guideline of 5 $\mu g\ m^{-3}$ (WHO, 2021). More specifically, $PM_{2.5}$ sensitivities in Euro_Medi are complex and vary from the north to south. The UK and Scandinavia are more sensitive to $NH_3$ emissions reductions, central Europe is more sensitive to $NO_x$, while the Mediterranean region is more sensitive to $SO_x$, which is consistent with conclusions from other European studies (Megaritis et al., 2013; Vieno et al., 2016; Aksoyoglu et al., 2020; Jiang et al., 2020; Clappier et al., 2021). From a perspective of Europe-wide policy making, it is important to reduce $NH_3$ and $NO_x$ emissions together and/or go for stronger reductions to minimise adverse effects caused by enhanced oxidation efficiency. In North America, the most effective measure to decrease annual mean $PM_{2.5}$ is reducing $NO_x$ emissions. Similar conclusions are reported by other studies. Liao et al. (2008) reported that the reduction in $NO_x$ emissions was most effective for decreasing 24-h mean $PM_{2.5}$ levels for five cities in the US, while Kelly et al. (2021) reported that reducing $NO_x$ emissions was more effective for reducing $PM_{2.5}$ concentrations in the eastern US than $SO_2$, $NH_3$, and VOC emissions reductions in both January and July. Tsimpidi et al. (2007, 2008) showed that $NO_x$ emissions reductions were the most effective measure for controlling $PM_{2.5}$ in the eastern US in summer due to the combined effects of lower atmospheric oxidant levels and smaller precursor emissions. Unfortunately, no conclusion on annual mean $PM_{2.5}$ sensitivities was drawn from these studies.

PM$_{2.5}$ in South Asia is most sensitive to SO$_x$ emissions reductions, and least sensitive to NH$_3$ emissions reductions, which is because South Asia is extremely ammonia-rich (Ge et al., 2022) so reducing NH$_3$ has little impact on mitigating SIA. In this study, 40% reductions in SO$_x$ and NH$_3$ emissions respectively produce 10% and 1% decreases in South Asia regional average PM$_{2.5}$. Pozzer et al. (2017) also showed that NH$_3$ emissions control has negligible impacts on PM$_{2.5}$ levels in South Asia with a 50% reduction in NH$_3$ emissions only reducing annual mean PM$_{2.5}$ by 2%. Given the continuing strong growth of SO$_2$

emissions in India now and in future projections (Sadavarte and Venkataraman, 2014; Hoesly et al., 2018; Aas et al., 2019; Szopa et al., 2021), it is anticipated that PM$_{2.5}$ levels in this region will continue to rise.

    For East Asia, although considerable emissions reductions in SO$_2$ have been taking place in China, and to a lesser extent reductions in NO$_x$ (Liu et al., 2016; Hoesly et al., 2018; Zheng et al., 2018), continuous efforts need to be put into reducing not only SO$_2$ and NO$_x$ but also NH$_3$ as reductions in the three precursors are equally effective for mitigating PM$_{2.5}$. Studies

focusing on East Asia also highlight the importance of NH$_3$ emissions control for reducing annual PM$_{2.5}$ pollution especially in central and eastern China (Wang et al., 2011; Pozzer et al., 2017; Cheng et al., 2021). In addition, due to the high proportion of SIA in PM$_{2.5}$ in East Asia, reducing all 3 precursors together also produces the greatest reduction in regional average PM$_{2.5}$ compared to South Asia, Europe, and North America. Another benefit of collective emissions reductions is that it minimises the adverse side effects from individual reductions since these disbenefits can cause transitory increases in PM$_{2.5}$ and deposition

of certain N$_r$ and S$_r$ species when emission reduction measures are gradually applied.

    In short, considering the evolution of the chemical state of the atmosphere in Europe and North America since the industrial revolution, the measures to combat PM$_{2.5}$ pollution in the two regions may be a good reference for countries in East and South Asia.

**4.4 Non-one-to-one linearity**

Analyses of both 20% and 40% emissions reductions reveal different kinds of linearity and non-linearity. There is considerable non-linearity in the sense of a lack of one-to-one proportionality between an emission reduction and a species concentration or deposition change. Precursor emissions reduction sometimes even increases other pollutant concentrations. This is consequent on interactions in SIA formation, atmospheric oxidizing capacity, and N and S deposition as discussed earlier, and is highly geographically variable. Such non-one-to-one proportionality may be more significant in a certain area

during a certain season due to the inter-annual variability in local meteorology and emissions profiles. However, whilst the sensitivity of an annual quantity of a pollutant to emissions reductions is subject to seasonality, we are confident that the broad conclusions of this study will hold, and from a global and regional policymaking perspective, it is more practical to develop policies of reductions in emissions from different sectors and countries on an annual level. Studies focusing on non-proportional responses of European PM$_{2.5}$ to emissions reductions showed that significant seasonality only occurs in a few

specific areas (Thunis et al., 2015, 2021; Clappier et al., 2021).

    On the other hand, a linearity in response to emissions reductions is apparent via the observation that the responses of PM$_{2.5}$, N$_r$, and S$_r$ annual concentrations and deposition components remain essentially proportional to the precursor emissions reductions (20% and 40%) for a given precursor in a given region, albeit that the magnitude of the slope varies substantially with different precursors and regions. Even if the net concentration changes of one species (e.g., PM$_{2.5}$) induced by reductions

in emissions of all 3 precursors are smaller (or greater) than the sum of changes from reductions in individual precursors due to different chemical interactions as discussed earlier, these net changes still follow a very similar gradient when emissions reductions in all 3 precursors change from 20% to 40%. However, where the gradient in the response is not one-to-one, the linearity in the response that is observed up to the 40% emissions reductions simulated here clearly cannot continue to extrapolate linearly all the way to 100% emissions reductions. The gradient of the response must be flatter or steeper

(depending on atmospheric component) at the beginning or end of the span from zero to 100% emissions reduction. For instance, 20% and 40% reductions in NH$_3$ emissions give 24% and 48% decreases in global annual mean NH$_3$ concentrations

respectively (i.e., gradients exceeding one-to-one), whereas these emissions reductions only produce 6% and 14% decreases in global $NH_4^+$ concentrations, respectively (i.e., gradients less than one-to-one). When $NH_3$ emissions are completely switched off, both $NH_3$ and $NH_4^+$ concentrations will be zero, so the $NH_3$ concentration sensitivity must become smaller, and the $NH_4^+$ concentration sensitivity must become larger, as $NH_3$ emissions reductions approach 100%. Additional serial sensitivity experiments are required to acquire the full spectrum of $NH_3$ and $NH_4^+$ (and all other species) sensitivities. It is therefore important for policy-makers in different regions to know the emissions reductions required to obtain the mitigation responses needed for specific air quality targets.

**4.5 Implications and limitations**

The parameterisation of atmospheric processes in ACTMs is in continuous interplay with the experimental evidence. The responses of $N_r$ and $S_r$ species to emissions reductions in $NH_3$, $NO_x$ and $SO_x$ as discussed in previous sections reveal several complex processes incorporated in the EMEP MSC-W model. The extent to which these parameterised processes accurately represent the reality of the atmosphere calls for more experimental evidence (field and/or laboratory measurements).

For instance, the synergistic interaction between $NH_3$ and $SO_2$ dry deposition to leaf surfaces is well known and has been termed 'co-deposition'. The idea is that the existence of both $NH_3$ and $SO_2$ acts to reduce the canopy resistances and increase the dry deposition rate for both gases (Sutton et al., 1994; Fowler et al., 2001; Nemitz et al., 2004; Fowler et al., 2009; Massad et al., 2010). This co-deposition effect is incorporated into the EMEP MSC-W model, making use of empirical parameters derived from European field measurements (Erisman et al., 2001) which may not be completely applicable to other world regions (e.g., South America). In addition, for regional variations in atmospheric oxidizing capacity caused by $NO_x$ emissions reductions, there are several studies reporting measurements evidence in East Asia, Europe, and North America, while no such measurements are found in South Asia. Long-term standardised measurements for both surface concentrations and wet and dry deposition with a sufficient amount of sampling sites in each geographical region are required for corroborating globally modelled results. The quantitative results reported in this study can serve as a reference for future modelling and measurement studies, albeit firmer field evidence worldwide is required to develop a reliable and robust chemistry and deposition schemes for global models.

Finally, it is also important to remember that reductions in anthropogenic emissions of SIA precursors will have many co-benefits on forest health, ecosystem biodiversity, and climate, not just in populated areas but elsewhere. For instance, $NH_3$ has become a major air pollution driver of lichen distributions in many European forests in recent years. In Scotland, Sutton et al. (2009) showed how lichens were gradually eradicated in areas near a poultry farm which is a large emitter of $NH_3$. Similarly, van Herk. (2001) reported that increased ambient $NH_3$ concentrations in the Netherlands appear to be the primary cause of the disappearance of acidophytic lichen species (i.e., species that prefer naturally acidic bark) over the last decade. Moreover, although the effects on the availability of nutrients in terrestrial and aquatic ecosystems are broadly assumed to be decided by total N inputs, Sutton et al. (2020) placed a stronger emphasis on the form in which the N was deposited. In their experiments, dry deposition of $NH_3$ showed a larger toxicity than wet deposition of $NH_4^+$ and $NO_3^-$. In this case, policy-makers should be more cautious about emissions controls with side effects of increased $NH_3$ dry deposition. Meanwhile, since most $NH_3$ is emitted via volatilization, a warmer atmosphere will promote its global emissions (Johnson et al. 2008; Sutton et al. 2013; Riddick et al. 2018). $NH_3$ emission controls thus need to include both direct reductions and indirect measures to abate climate warming as well.

## 5 Conclusions

The sensitivities of global and regional annual mean surface concentrations and deposition of gaseous and particle $N_r$ and $S_r$ to 20% and 40% reductions in anthropogenic emissions of $NH_3$, $NO_x$, and $SO_x$ both individually and collectively has been investigated using the EMEP MSC-W model coupled with WRF meteorology for 2015. East Asia, South Asia, Euro_Medi, and North America are selected for regional discussions because of their high population densities and $N_r$ and $S_r$ pollution, and because the model outputs are evaluated and agree reasonably well with measurements in most of these areas. The comparison in regional responses reveals that the emissions reduction in one precursor has multiple co-benefits and sometimes small disbenefits on mitigating $N_r$, $S_r$, and $PM_{2.5}$ pollution, and these effects are highly geographically variable.

Whilst reductions in $NH_3$ emissions are effective for decreasing annual $NH_3$ concentrations and deposition they are considerably less effective at decreasing $NH_4^+$. This is because all densely populated continents are ammonia-rich so reducing $NH_3$ emissions only has limited effects on mitigating SIA formation. A 40% reduction in $NH_3$ emissions decreases regional average $NH_3$ concentrations in the four regions by 47-49%, while $NH_4^+$ concentrations decrease in the order Euro_Medi (18%), East Asia (15%), North America (12%), and South Asia (4%), the order of increasing regional ammonia-richness. A notable disbenefit is increased $SO_2$ concentrations because reduced $NH_3$ levels affect the pH-dependent $SO_2$ dry deposition. A 40% reduction in $NH_3$ emissions increases $SO_2$ concentrations in East Asia by 16%, in South Asia and North America by 14%, and in Euro_Medi by 10%.

Large regional differences are observed in $NO_x$ emissions reduction scenarios. In East Asia, $NO_x$ concentrations are very effectively decreased (by 45%) with 40% $NO_x$ emissions reductions, but they are less effectively decreased in Euro_Medi (38%) and North America (36%), and to a least extent in South Asia (22%). By contrast, the regional sensitivities of fine $NO_3^-$ are reversed: South Asia shows the largest decrease (45%), whilst East Asia shows the smallest decrease (33%). This phenomenon is related to different regional oxidation regime and SIA chemistry. $NO_x$ emissions reductions increase $O_3$ levels in East Asia (and also, but by less, in Euro_Medi), but decrease $O_3$ levels in South Asia (and also, but by less, in North America), which causes an enhanced $NO_x$ and $SO_x$ oxidation in former regions but a decreased one in latter regions. Consequently, increased $SO_4^{2-}$ and $SO_2$ concentrations appear in East Asia and South Asia respectively.

Reductions in $SO_x$ emissions have globally consistent impacts on $SO_2$ and $SO_4^{2-}$ concentrations. A 40% reduction in $SO_x$ emissions decreases $SO_2$ and $SO_4^{2-}$ concentrations in the four regions by 42-45% and 34-38% respectively, while the disbenefit is that decreased $(NH_4)_2SO_4$ formation yields ~12% growth in $NH_3$ total deposition.

This work also highlights important messages for policy-makers concerning the mitigation of $PM_{2.5}$. More emissions controls focusing on $NH_3$ and $NO_x$ are necessary for regions with better air quality such as Europe and North America. In Euro_Medi, $PM_{2.5}$ sensitivities vary from the north to south, with $NH_3$ reductions being more effective for UK and Scandinavia, $NO_x$ for central Europe, and $SO_x$ for the Mediterranean. In South Asia, $PM_{2.5}$ is most sensitive to $SO_x$, and least sensitive to $NH_3$, which is consistent with the fact that South Asia is so ammonia-rich that reducing $NH_3$ hardly has any impacts. Given the continuing strong growth of $SO_2$ emissions in India now and in future projections, it is anticipated that $PM_{2.5}$ levels in this region will continue to rise. In East Asia, although considerable emissions reductions in $SO_2$ have been taking place in China, and to a lesser extent reductions in $NO_x$ also, continuing efforts need to be put into reducing not only $SO_2$ and $NO_x$ but also $NH_3$ as reductions in the three precursors are equally effective for mitigating $PM_{2.5}$.

This work reveals some geographically-varying and non-one-to-one proportionality of chemical responses of $N_r$, $S_r$, and $PM_{2.5}$ to emissions reductions. It is thus important not only to prioritise different emission controls in different regions, but also to reduce several emissions together in order to minimise the potential disbenefits.

**Code and data availability**

As described and referenced in Sect. 2 of this paper, this study used two open-source global models: the European Monitoring and Evaluation Programme Meteorological Synthesizing Centre – West atmospheric chemistry transport model (EMEP MSC-W, 2020, version 4.34, source code available at https://doi.org/10.5281/zenodo.3647990, last access: 18 January 2023) and the Weather Research and Forecasting meteorological model (WRF, version 3.9.1.1, https://www.wrf-model.org, last access: 8 Aug 2022; Skamarock et al., 2008). The model outputs presented in figures and tables in this paper and the corresponding

Python scripts are available at https://doi.org/10.5281/zenodo.7082661, last access: 18 January 2023 (Ge, 2022).

**Author contribution**

MH, DS and MV conceptualised and supervised the study. MV and PW contributed to model development and set-up and provided modelling support. MV provided computing resource. YG contributed to study design, undertook all model simulations, formal data analyses, visualisation of the results and data curation, with discussion and refinement by all authors.

The original draft of the paper was written by YG with contributions and editing by MH. All authors provided review comments and approval of the final version.

**Competing interests**

The authors declare that they have no conflict of interest.

**Acknowledgments**

Y. Ge gratefully acknowledges studentship funding from the University of Edinburgh and its School of Chemistry. This work was in part supported by the UK Natural Environment Research Council (NERC), including grant nos. NE/R016429/1 and NE/R000131/1, the Department for Environment, Food and Rural Affairs (Defra) contract "Research & Development Support for National Air Pollution Control Strategies (ECM: 62041) 2021 to 2024", and the European Modelling and Evaluation Programme under the United Nations Economic Commission for Europe Convention on Long-range Transboundary Air

Pollution.

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
