# Peer review of "Global sensitivities of reactive N and S gas and particle concentrations and deposition to precursor emissions reductions"

_Atmospheric Chemistry and Physics, 2022_

## Author Comment (AC1)

**acp-2022-657: Global sensitivities of reactive N and S gas and particle concentrations and deposition to precursor emissions reductions**
Ge et al.

**Response to Reviewer #1**

We thank the reviewer for their time spent reading our manuscript and for their recommendation for publication upon addressing their comments. Below we include all the reviewer's comments and provide in blue text our point-by-point responses. Please note that the line numbers mentioned in our responses refer to the clean revised manuscript (not the track-changed version).

**General comments 1:**
The paper systematically describes an evaluation of PM and N/S concentrations and deposition sensitivity to precursor emissions, but overall, I feel a too weak connection with any empirical evidence:

1. Although the paper is already very long, I suggest adding a brief section and 1-2 new figures that describe a "base" scenario that shows the simulated concentrations of related chemical species compared against available observations.

Response: We have already published in Ge et al. (2021) a comprehensive evaluation of surface concentrations and wet deposition of $N_r$ and $S_r$ species from this model configuration for 2010 and 2015 against global measurements from 10 monitoring networks. This paper demonstrated the model's capability to capture the overall spatial variations in annual concentrations of $NH_3$, $NH_4^+$, $NO_2$, $HNO_3$, fine $NO_3^-$, $SO_2$ and $SO_4^{2-}$ and their wet deposition in East Asia, Southeast Asia, Europe, and North America, which supports the application of this model framework for global and regional sensitivity analyses in this study.

Also, in another publication, Ge et al. (2022), we already investigated the current chemical state for $PM_{2.5}$ formation in our 'base' scenario. The paper highlights important atmospheric processes controlling $N_r$ and $S_r$ regional distributions, which provides a basis for the explanations of the responses of $N_r$ and $S_r$ species to potential emission controls as discussed in this study.

Together, our two previous papers provide a comprehensive overview of the baseline model-measurement comparison and global budgets, which is why we did not present details of these analyses again in the current paper but provided citation to them at appropriate places (e.g., lines 122-129 in the main paper for Ge et al. (2021)). Given that our current paper is already

very long, we do not want to add material that is already published but instead have added the following brief introduction of the baseline scenario as a Sect. S1 of the Supplement:

"The global model evaluation of $N_r$ and $S_r$ concentrations and wet deposition from this model configuration for 2010 and 2015 against measurements from 10 ambient monitoring networks is documented in Ge et al. (2021) and demonstrates the model's capability for capturing the spatial and seasonal variations of $NH_3$, $NH_4^+$, $NO_2$, $HNO_3$, $NO_3^-$, $SO_2$, and $SO_4^{2-}$ in East Asia, Southeast Asia, Europe, and North America. Figure S1 gives an example of model-measurement comparisons of 2015 annual average surface concentrations of $NH_3$ and $NH_4^+$, and annual wet deposition of reduced N (RDN = $NH_3$ + $NH_4^+$) in East Asia, Southeast Asia, Europe, and North America. Model and measurements consistently show higher RDN concentrations and wet deposition in East Asia compared to other regions, which is consistent with East Asia becoming a hot spot of RDN pollution in recent years (Szopa et al., 2021; Hoesly et al., 2018). The modelled annual average $NH_3$ concentrations show similar agreement with measurements from the four regions, with the correlation coefficient $R$ ranging from 0.56 to 0.72. The linear correlations between modelled and measured $NH_4^+$ are highest in Southeast Asia, followed by Europe and North America, while East Asia shows a relatively poor correlation, reflecting potential differences among individual measurement networks. For wet deposition of RDN, the model simulates smaller values by 21% - 50% across 5 different networks. Further examination of wet deposition components reveals that this is largely driven by a general underestimation of annual total precipitation (Ge et al., 2021). Given the localised nature of precipitation events and the intrinsic scale mismatch between a 1° model grid volume average and a single sampling point, such a model underestimation range is expected.

The model-measurement comparison metrics in this work are comparable with other global modelling studies. Hauglustaine et al. (2014) reported that the $R$ values of their global model results (LMDz-INCA global chemistry–aerosol–climate model, 1.9° latitude × 3.75° longitude resolution) versus measurements in 2006 for surface concentrations of $SO_4^{2-}$, $NH_4^+$ and $NO_3^-$ ranged 0.43-0.58 in Europe and 0.54-0.77 in North America, which is similar to our results presented here. The AeroCom phase III global nitrate experiment, which includes 9 models, reported slightly lower $R$ ranges than here for annual $NO_3^-$ in 2008: 0.081-0.735 in North America, 0.393-0.585 in Europe, and 0.226-0.429 in Southeast Asia (Bian et al., 2017). Again, for detailed analyses of evaluation statistics of other species, please refer to our previous model evaluation study Ge et al. (2021)."

[Figure]

**Figure S1. Comparisons of 2015 annual average surface concentrations of NH₃ (top row) and NH₄⁺ (middle row), and annual wet deposition of reduced N (RDN = NH₃ + NH₄⁺; bottom row) between model and measurements in East Asia (Chinese NNDMN network), Southeast Asia (EANET network), Europe (EMEP network) and North America (US EPA and Canadian NAPS networks). In each plot, R is the Pearson correlation coefficient, the solid line is the least-squares regression line, and the dashed black line is the 1:1 line. Detailed information about measurement networks is presented in Ge et al. (2021).**

2. Such a 20-40% emission reduction occurred in many parts of the world, for example, the United States/Canada, EMEP, and East Asia regions. There are also long-term air quality observations available in these regions. How are your sensitivity results compared with this observational evidence?

Response: The sensitivity results in this study cannot be directly compared with observational evidence, since the real-world emission changes involve multiple species increasing/decreasing to a different extent at the same time, and also differently in different locations. Additionally, it is impossible to quantify from measurements the sensitivity of individual deposition components to emissions changes (e.g., the extent to which wet deposition of reduced N is driven by sensitivity of rainout of NH₄⁺ or rainout of gaseous NH₃). This also limits direct comparison of model sensitivities to observations.

However, we did already undertake a similar analysis in Ge et al. (2021) in which we compared the modelled concentration and total deposition responses to changes in emissions between 2010 and 2015 with the measurement data. These comparisons of model and measurement responses between the two years provide useful additional confirmation that the model responses are in line with expectations, subject to the following two caveats. First, the need for there to be measurement sites operating in both 2010 and 2015 substantially reduces the number of comparisons between the two years for some measurement networks and some species. Secondly, the comparison of the model (and measurement) changes between the two years with the emissions changes between the two years is confounded by any change in relevant meteorology between the two years. The results of this analysis consisting of an individual section of text, 4 figures and a table are reported in the Supplement (Pages 8-16) of Ge et al. (2021). For information, we provide here an excerpt from that section in our previous paper:

"Table S1 *(N.B. not reproduced in these responses)* shows that modelled concentration changes between 2010 and 2015 are highly consistent with the measured trends (and with the trends in emissions) for all three of the precursor gases $NH_3$, $SO_2$ and $NO_2$. For example, the annual average emissions and measured and modelled concentrations of $SO_2$ for all networks with $SO_2$ data (EANET, UK, EMEP, Canada, and US) all show clear decreasing trends from 2010 to 2015 with relative decreases generally in the range of −20% to −40%. Similarly, there is consistency between model and measurements (and with emissions) that there is no trend in $NH_3$ between 2010 and 2015 for all networks except the China network for which there is again consistency between modelled and measured concentrations that $NH_3$ increased between these two years in this region, as do the emissions albeit by a smaller relative amount. (There is also indication of small upward trend in $NH_3$ in the US network). For $NO_2$, there is again very good consistency of relative trends between model and measured concentrations of no change in USA, Canada and China, and a decrease in Europe. For the EANET sites the model simulates a somewhat larger decrease in $NO_2$ concentrations (−22%) than shown by the measurements (−6.4%) but this comparison is only based on 7 data points.

The responses of secondary species to emission changes from 2010 to 2015 are more complex with clear upward and downward trends as well as no trend appearing in separate networks. For $SO_4^{2-}$ concentrations, as for $SO_2$ concentrations, both model and measurements are consistent in showing strong decreases between 2010 and 2015 across all the network sites except for EANET for which the model simulates a modest decrease not shown in the measurements. $NH_4^+$ concentrations show both clear positive and negative changes in separate networks. $NH_4^+$ measurements in China and EANET networks both increase by 18%, whereas modelled $NH_4^+$ concentrations both show no obvious trend between 2010 and 2015. For networks in UK and Europe (EMEP/CCC), modelled and measured $NH_4^+$ concentrations show similar downward trends all ranging around −21%. $NH_4^+$ measurements in US and Canada networks decrease by −39% and −26% respectively, whilst their corresponding modelling data

shows smaller decreases. For $HNO_3$ and $NO_3^-$, a mixture of upward trend, downward trend, and no trend appears in separate networks and in both model and measurements as well, reflecting a varying response mechanism. In general, however, the 2010 to 2015 relative trends (including no trend) for $NH_3$, $NH_4^+$, $NO_2$, $HNO_3$, and $NO_3^-$ are consistent between model and measurements."

**General comments 2:**

The abstract and conclusion sections need to be significantly shortened, less repetitive, and with a higher-level summary. The results section is generally too long, while the discussion section is too short and weak. The discussion should be expanded significantly with subsections. The research implication of the article is not emphasized enough, giving the readers the impression that it is a purely technical report. Besides, the discussion on the sources of uncertainty needs to be elaborated in more detail.

Response: In the revised manuscript, we have shortened both the Abstract and Conclusion sections as requested.

The results section is indeed quite long, but it presents a comprehensive analysis of how different $N_r$ and $S_r$ species interact with each other and how that affects $PM_{2.5}$ composition and the nature of how N and S deposits in different regions. Considering the complexity of these interactions and their regional differences, we would like to retain this section in its current form.

For the discussion section, we have expanded it with subsections. More information is added to the discussion of model uncertainty as Sect. 4.1. We draw attention that we also discussed model uncertainty in our previous paper, Ge et al. (2021). The take-home messages and the implications of this work are further emphasized with more details in Sect. 4.2, 4.3 and 4.5. Due to the large number of edits, we have made to address this reviewer comment, it is not practical to copy and paste every change into this response document so please refer to our revised manuscript for these revised subsections.

**General comments 3:**

The equation to calculate the sensitivity and the definition of sensitive areas need to be described in more detail within the method section. Why are only these four regions analyzed? Why not include more regions in the world, such as Middle/South America, Africa, and the Middle East? These regions may have sensitive ecosystems, emerging economic growth, and also large populations. The omission of these regions needs better reasoning.

Response: We have now added the equations to calculate the sensitivities to Sect. 2.2 (lines 143-147) in the revised manuscript as follows:

"The sensitivity ($Sensitivity_i$) of the concentration/deposition of a species $i$ is calculated as the absolute difference between the value in baseline ($Baseline_i$) and in an emission reduction scenario ($Scenario_i$). Taking $NH_3$ concentration as an example:

$$Sensitivity_{NH_3}(\mu g\ m^{-3}) = Baseline_{NH_3} - Scenario_{NH_3}$$

For the relative sensitivity ($Relative\ Sensitivity_i$):

$$Relative\ Sensitivity_{NH_3}(\%) = \frac{Sensitivity_{NH_3}}{Baseline_{NH_3}} \times 100\% \quad "$$

Our discussion focuses on East Asia, South Asia, Euro_Medi, and North America on account of both the thoroughly evaluated model results and the high population density and high $N_r$ and $S_r$ pollution in those regions. For regions like Central/South America, Africa, and the Middle East, we were unable to conduct model-measurement comparisons because no measurement data for the modelled years are publicly available. By comparison, we have more confidence in model outputs in regions where the model is comprehensively evaluated and shown to behave reasonably well. We have now added the following sentences to Sect. 2.3 (lines 156-159) to provide a more detailed explanation:

"All four regions are densely populated and have high $N_r$ and $S_r$ pollution. Besides, due to limitations in the number of publicly available measurements, our model outputs are evaluated against measurements in East Asia, Europe, and North America, and therefore we have greater confidence in sensitivity results in these three regions. South Asia is chosen because of its extreme ammonia richness, as revealed by Ge et al. (2022), which makes it an interesting comparison with other regions."

**Specific comments:**

1. Line 49-50. There is a grammatical error, consider changing "also its form" to "also by its form".

Response: Requested change made (now in line 43).

2. Line 57-58. This sentence is vague. It is not enough to support your point on emphasizing the global dominance of emissions in East and South Asia.

Response: We thank the reviewer for pointing out the vagueness of this sentence. It is now rephrased as follows (lines 49-52):

"Historically, Europe and North America were the dominant emissions regions, suffering severe air pollution until the late 20[th] century. As reductions in $SO_x$ and $NO_x$ emissions took

effect in Europe and North America, emissions in East and South Asia increased dramatically due to rapid industrialisation and dominated global $N_r$ and $S_r$ emissions by the early 21st century"

3. Line 89-95. The description of the single reference cited here is too long, you may consider summarizing it.

Response: The description of this reference is shortened as follows (lines 83-86):

"Holt et al. (2015) used GEOS-Chem to investigate $PM_{2.5}$ sensitivities in the United States to emissions reductions between two sets of scenarios representing a 2005 baseline (high emissions) and a 2012 analogue (low emissions). They found larger sensitivities of $PM_{2.5}$ to $SO_x$ and $NO_x$ controls in the low emissions case since lower $NO_x$ emissions in 2012 enhance the relative importance of aqueous-phase $SO_2$ oxidation."

4. Line 99-100. Consider changing "global" to "the global".

Response: Requested change made (now in line 90).

5. Line 175-178. These sentences may be moved to the discussion section.

Response: We prefer to keep these sentences in their current location because we feel this is an important point to make at this point and because Sect. 4.2 of the discussion already includes further elaboration on this.

6. Line 367. The method of the specific definition of sensitivity regime through the precursor includes sensitivities of $NH_3$, $NO_x$, and $SO_x$, which could be elaborated in your method section.

Response: An introduction of the definition of $PM_{2.5}$ sensitivity regimes has been added to Sect. 2.2, immediately after the introduction of sensitivity calculation paragraphs (lines 148-151):

"The sensitivities of different species are calculated for all emission reduction scenarios. The $PM_{2.5}$ sensitivities derived from individual reductions in emissions of $NH_3$, $NO_x$, or $SO_x$ are used to define the sensitivity regimes for different regions in Sect. 3.2. For each model grid, the regime is decided by the precursor that yields the greatest decrease in grid $PM_{2.5}$ concentration: $NH_3$ sensitive, $NO_x$ sensitive, or $SO_x$ sensitive."

7.  Line 382. It seems a bit obscure to be a separate paragraph here, are you trying to explain the sensitivity of $SO_x$ in the marine area?

Response: We would like to emphasize that although $PM_{2.5}$ concentrations in South Asia and marine areas are both more sensitive to $SO_x$ emission reductions, there are different reasons. In South Asia, it is because ammonium sulfate dominates SIA and this region is very ammonia-rich, so the availability of $SO_x$ is the limiting factor for SIA formation. In marine areas, however, anthropogenic emissions of $NO_x$ and $NH_3$ are very low, and sulfate aerosol derived from oceanic emissions of DMS is the major contributor to SIA. Therefore, reductions in $SO_x$ emissions have some effect on marine $PM_{2.5}$, but $NO_x$ and $NH_3$ reductions have almost no effect whatsoever. To make the message clearer, these sentences are not now placed in a separate paragraph and are rephrased as follows (lines 387-391):

"Furthermore, many marine areas are characterised as $SO_x$ sensitive but for a different reason than the $SO_x$ sensitivity in South Asia. In the marine areas, fine nitrate and ammonium aerosols are relatively small compared to sulfate aerosols, therefore reductions in $NO_x$ and $NH_3$ emissions hardly affect SIA formation. In fact, sulfate aerosol derived from oceanic emissions of DMS rather than from anthropogenic emissions is the major contributor to marine $PM_{2.5}$ (Quinn and Bates, 2011; Hoffmann et al., 2016; Novak et al., 2022)"

8.  Line 514-515. This sentence may be not proper here, please consider moving it to the discussion section.

Response: We prefer to keep this sentence in its current position as it provides a brief explanation for decreases in $SO_2$ dry deposition caused by $NH_3$ emissions reductions. More details are then given in the discussion section.

9.  Line 717. Is it possible that the significant seasonality in these few areas is related to the vegetation on the land surface?

Response: In Europe, Po basin (Italy) is the area that shows the largest seasonality in SIA responses to $NO_x$ emissions reductions (Thunis et al. 2021; Clappier et al. 2021). For instance, in Bergamo (a city in the Po basin), SIA decreases in summer but increases in winter, in response to 50% $NO_x$ emissions reductions. This is mainly driven by the seasonality in meteorology and emissions rather than the vegetation on the land surface. The increase in the inorganic fraction of $PM_{2.5}$ during wintertime has been related to an increase in the oxidising capacity of the atmosphere and in particular to increased levels of $O_3$ which is due to the reduction in the titration of $O_3$ by NO in wintertime high-$NO_x$ conditions in this region. By comparison, $NO_x$ levels in summer are relatively lower and this peculiar increase in SIA is not observed.

10. Line 737-738. Consider elaborating and expanding it by citing some related references.

Response: These sentences are expanded with more references as requested. In the revised manuscript, the new paragraph is presented as follows (lines 799-811):

"Finally, it is also important to remember that reductions in anthropogenic emissions of SIA precursors will have many co-benefits on forest health, ecosystem biodiversity, and climate, not just in populated areas but elsewhere. For instance, $NH_3$ has become a major air pollution driver of lichen distributions in many European forests in recent years. In Scotland, Sutton et al. (2009) showed how lichens were gradually eradicated in areas near a poultry farm which is a large emitter of $NH_3$. Similarly, van Herk. (2001) reported that increased ambient $NH_3$ concentrations in the Netherlands appear to be the primary cause of the disappearance of acidophytic lichen species (i.e., species that prefer naturally acidic bark) over the last decade. Moreover, although the effects on the availability of nutrients in terrestrial and aquatic ecosystems are broadly assumed to be decided by total N inputs, Sutton et al. (2020) placed a stronger emphasis on the form in which the N was deposited. In their experiments, dry deposition of $NH_3$ showed a larger toxicity than wet deposition of $NH_4^+$ and $NO_3^-$. In this case, policy-makers should be more cautious about emissions controls with side effects of increased $NH_3$ dry deposition. Meanwhile, since most $NH_3$ is emitted via volatilization, a warmer atmosphere will promote its global emissions (Johnson et al. 2008; Sutton et al. 2013; Riddick et al. 2018). $NH_3$ emission controls thus need to include both direct reductions and indirect measures to abate climate warming as well."

11. Line 778-779. This sentence seems a bit vague. Please check it.

Response: The conclusion section has been revised to make this message clearer.

12. Line 789-790. There are several grammatical errors here. Please check and rewrite it.

Response: Requested change made (now in lines 850-851).

**References**

Bian, H., Chin, M., Hauglustaine, D. A., Schulz, M., Myhre, G., Bauer, S. E., Lund, M. T., Karydis, V. A., Kucsera, T. L., Pan, X., Pozzer, A., Skeie, R. B., Steenrod, S. D., Sudo, K., Tsigaridis, K., Tsimpidi, A. P., and Tsyro, S. G.: Investigation of global particulate nitrate from the AeroCom phase III experiment, Atmospheric Chemistry and Physics, 17, 12911-12940, 10.5194/acp-17-12911-2017, 2017.

Ge, Y., Heal, M. R., Stevenson, D. S., Wind, P., and Vieno, M.: Evaluation of global EMEP MSC-W (rv4.34) WRF (v3.9.1.1) model surface concentrations and wet deposition of reactive N and S with measurements, Geosci. Model Dev., 14, 7021-7046, 10.5194/gmd-14-7021-2021, 2021.

Ge, Y., Vieno, M., Stevenson, D. S., Wind, P., and Heal, M. R.: A new assessment of global and regional budgets, fluxes, and lifetimes of atmospheric reactive N and S gases and aerosols, Atmos. Chem. Phys., 22, 8343-8368, 10.5194/acp-22-8343-2022, 2022.

Hauglustaine, D. A., Balkanski, Y., and Schulz, M.: A global model simulation of present and future nitrate aerosols and their direct radiative forcing of climate, Atmospheric Chemistry and Physics, 14, 11031-11063, 10.5194/acp-14-11031-2014, 2014.

Hoesly, R. M., Smith, S. J., Feng, L., Klimont, Z., Janssens-Maenhout, G., Pitkanen, T., Seibert, J. J., Vu, L., Andres, R. J., Bolt, R. M., Bond, T. C., Dawidowski, L., Kholod, N., Kurokawa, J. I., Li, M., Liu, L., Lu, Z., Moura, M. C. P., O'Rourke, P. R., and Zhang, Q.: Historical (1750–2014) anthropogenic emissions of reactive gases and aerosols from the Community Emissions Data System (CEDS), Geosci. Model Dev., 11, 369-408, 10.5194/gmd-11-369-2018, 2018.

Szopa, S., Naik V., Adhikary B., Artaxo P., Berntsen T., Collins W.D., Fuzzi S., Gallardo L., Kiendler-Scharr A., Klimont Z., Liao H., Unger N., and P., Z.: Short-Lived Climate Forcers. In Climate Change 2021: The Physical Science Basis. Contribution of Working Group I to the Sixth Assessment Report of the Intergovernmental Panel on Climate Change, Cambridge University Press, Cambridge, United Kingdom and New York, NY, USA, 817-922, 10.1017/9781009157896.008, 2021.

---

## Author Comment (AC2)

**acp-2022-657: Global sensitivities of reactive N and S gas and particle concentrations and deposition to precursor emissions reductions**
**Ge et al.**

**Response to Reviewer #2**

We thank the reviewer for their time spent reading our manuscript and for their recommendation for publication upon addressing their comments. Below we include all the reviewer's comments and provide in blue text our point-by-point responses. Please note that the line numbers mentioned in our responses refer to the clean revised manuscript (not the track-changed version).

**Specific comments 1: Model set-up**
Global 20% and 40% emission reductions were applied in this study. Some sentences are needed to justify the selection of 20% and 40%. Why a greater emission reduction (i.e., 60%) was not considered?

Response: We thank the reviewer for their suggestion. In this work, the design of modelling experiments involves balancing two factors. On one hand, the number of simulations is constrained by available computing resources and storage space. On the other hand, we aim to consider reductions that are both realistic and achievable, such as 20% and 40%, without being overly ambitious for certain regions. However, as discussed in the Discussion section, the non-one-to-one but relatively linear responses of $N_r$ and $S_r$ concentrations and deposition to 20% and 40% emissions reductions suggest that more ambitious reductions may be needed in the future.

In the revised manuscript, the following sentence (lines 139-141) is revised to justify our selection of reduction levels:

"Limited by available computational resources and storage space and taking the achievability of real-world emissions controls into account, the model experiments applied 20% and 40% reductions to global anthropogenic emissions of $NH_3$, $NO_x$, $SO_x$ from all sectors both individually and collectively (i.e., reductions applied to all 3 species simultaneously)."

**Specific comments 2: Section 3.1.2, Line 260**
What are the sources of fine and coarse nitrate aerosol? Can you explain why coarse nitrate would increase associated with $NH_3$ emission reduction? Please clarify.

Response: The fine nitrate aerosol in EMEP MSC-W model is essentially $NH_4NO_3$ which is produced from the reaction between $HNO_3$ and $NH_3$. The coarse nitrate comes from reactions between $HNO_3$ and coarse particles (e.g., dust and sea salt). Reductions in $NH_3$ emissions cause the equilibrium between $HNO_3$ and $NH_3$ to shift away from $NH_4NO_3$ production and therefore free more $HNO_3$. As a result, more $HNO_3$ is available to produce coarse nitrate aerosol. In the revised manuscript, we revised this sentence to provide a clearer explanation (lines 263-265):

"Reductions in $NH_3$ emissions cause the equilibrium between $HNO_3$ and $NH_3$ to shift away from $NH_4NO_3$ production and therefore free more $HNO_3$ molecules. As a result, more $HNO_3$ is available to produce coarse nitrate aerosol, leading to a decrease in fine $NO_3^-$ but an increase in coarse $NO_3^-$ concentrations (Fig. S2)."

**Specific comments 3: Page 18, Figure 8**
Figure 8 showed the spatial sensitivity regimes based on 40% emission reductions and annual mean PM2.5 concentrations. Such sensitivity regimes shall have large seasonal variations. Do you have the model datasets to generate the seasonal maps? That shall provide valuable information to understand the SIA formation regimes.

Response: We do have monthly model outputs to generate seasonal maps. We calculated seasonal sensitivity regimes for the globe and noticed some seasonal variations in $PM_{2.5}$ sensitivity regimes in East Asia and Europe, which indeed reveals some interesting subtleties regarding the limiting factor in SIA formation in different regions. Considering this manuscript is already very long, we put this seasonal analysis into the supplement as Sect. S2:

"To reveal more details of temporal variations in global $PM_{2.5}$ sensitivity regimes, we compare $PM_{2.5}$ sensitivities to individual emissions reductions on a seasonal basis using the Northern Hemisphere calendar: spring (March, April, and May), summer (June, July, and August), autumn (September, October, and November), and winter (December, January, and February). Figure S7 presents the spatial distribution of dominant $PM_{2.5}$ sensitivity regimes in four seasons. In East Asia, the dominant regime shifts from $NO_x$-sensitive to $SO_x$-sensitive from spring to summer, while the $NH_3$-sensitive regime expands more and more from autumn to winter. Similar trends are observed across Europe as well, where $NO_x$-sensitive grids are prevalent during spring while $NH_3$-sensitive grids dominate during winter. The springtime $NO_x$-sensitive regime in these regions can be attributed to large $NH_3$ emissions from intensive agricultural activities in this season (Cheng et al., 2021; Dammers et al., 2019), which leads to the formation of $NH_4NO_3$ being primarily limited by the availability of $HNO_3$. Consequently, reductions in $NO_x$ emissions decrease gaseous $HNO_3$ production which then decrease SIA concentrations. In the summer, $NH_4NO_3$ becomes less stable due to the generally higher temperature and sulfate aerosols remain a significant contributor to $PM_{2.5}$ in East Asia (Ianniello et al., 2011; Wang et al., 2013). Since the production of sulfate aerosols depends on the oxidation processes of $SO_2$ rather than the availability of $NH_3$, and $NH_3$ is in excess anyway, $SO_x$ emissions

reductions become the most effective single-precursor control for $PM_{2.5}$ mitigation in this region. The wintertime $NH_3$-sensitive regime in both Europe and East Asia is caused by smaller $NH_3$ emissions (due to reduced agricultural activities) and relatively larger $NO_x$ emissions (such as from increased domestic heating). Changes in meteorological factors (e.g., decreased vertical dispersion) may also contribute to higher $NO_x$ surface concentrations in the winter. As a result, $NH_3$ becomes the limiting factor in $NH_4NO_3$ formation and therefore has the greatest impact on $PM_{2.5}$ sensitivities.

In contrast, North America and South Asia do not show significant seasonal variations in $PM_{2.5}$ sensitivity regimes. In the eastern US, $PM_{2.5}$ formation is $NO_x$-sensitive for most of the year, except for the summer when it is $SO_x$ sensitive. This suggests that further reductions in $NO_x$ emissions are necessary to decrease annual $PM_{2.5}$ levels in this region. In South Asia, the $SO_x$-sensitive regime dominates throughout the year, with the exception of northern India in the winter, which is more $NO_x$-sensitive. As discussed in the main paper, the extreme $NH_3$-richness and dominant contribution of sulfate aerosols to SIA in South Asia render $PM_{2.5}$ formation almost exclusively sensitive to $SO_x$ emission reductions.

[Figure]

Figure S7: Spatial and seasonal variation in sensitivity regime of $PM_{2.5}$ mitigation based on data from 40% individual reductions in emissions of $NH_3$, $NO_x$, or $SO_x$. The regime is defined according to the precursor that yields the greatest decreases in grid seasonal average $PM_{2.5}$ concentration: $NH_3$ sensitive (yellow), $NO_x$ sensitive (blue), $SO_x$ sensitive (green). Model grids with baseline seasonal mean $PM_{2.5}$ concentrations $<5$ $\mu g$ $m^{-3}$ are masked out."

**Specific comments 4:**

For the green/red circles and stars in Figures 2, 4, 6, 9, and 11, in the main text, the symbols that are discussed as the maximum reductions (e.g., Page 11, Line 330-335) were denoted as "Min" in these figures. Please be consistent.

Response: The reason for denoting the maximum reduction as "Min" is because the actual differences are negative values. The "Max" and "Min" points in these figures represent the maximum and minimum differences between baseline and emissions reduction scenarios respectively, so if these differences are positive values, they are described as increases (rather than decreases) in the main text. Therefore, we would like to retain these figures in their current form.

**Specific comments 5: Page 25, Line 708-710**

It is not clear what "non-linearity" mean here in the text. We can see from Figures 3, 5, 7, and 10, the responses with respect to 20% vs. 40% emission reductions are rather linear. They deviate from the 1:1 line, however, the responses are linear. Please clarify.

Response: We thank the reviewer for pointing out this ambiguity. In the revised manuscript Sect. 4.4 (lines 752-781), we rephrased the discussion of "non-linearity" to "non-one-to-one linearity". The non-linearity is defined as a lack of one-to-one proportionality between an emission reduction and a species concentration or deposition change. Meanwhile, it is also recognised in the same section that a linearity in response to emissions reductions is apparent via the observation that the responses of $PM_{2.5}$, $N_r$, and $S_r$ annual concentrations and deposition components remain essentially proportional to the precursor emissions reductions (20% and 40%) for a given precursor in a given region, albeit that the magnitude of the slope varies substantially with different precursors and regions.

**Specific comments 6:**

The writing of the manuscript is rather intensive. Many results are described in parallel, which makes the manuscript less focused. I understand that many results can be derived from the set of sensitivity simulations, still, the key findings of the study shall be better emphasized in the abstract and conclusions.

Response: We agree with the reviewer that the abstract and conclusion are too long. In the revised manuscript, we have shortened the two sections as requested to deliver a more focused message.

**References**

Cheng, L., Ye, Z., Cheng, S., and Guo, X.: Agricultural ammonia emissions and its impact on PM2.5 concentrations in the Beijing–Tianjin–Hebei region from 2000 to 2018, Environmental Pollution, 291, 118162, https://doi.org/10.1016/j.envpol.2021.118162, 2021.

Dammers, E., McLinden, C. A., Griffin, D., Shephard, M. W., Van Der Graaf, S., Lutsch, E., Schaap, M., Gainairu-Matz, Y., Fioletov, V., Van Damme, M., Whitburn, S., Clarisse, L., Cady-Pereira, K., Clerbaux, C., Coheur, P. F., and Erisman, J. W.: NH3 emissions from large point sources derived from CrIS and IASI satellite observations, Atmos. Chem. Phys., 19, 12261-12293, 10.5194/acp-19-12261-2019, 2019.

Ianniello, A., Spataro, F., Esposito, G., Allegrini, I., Hu, M., and Zhu, T.: Chemical characteristics of inorganic ammonium salts in PM2.5 in the atmosphere of Beijing (China), Atmos. Chem. Phys., 11, 10803-10822, 10.5194/acp-11-10803-2011, 2011.

Wang, Y., Zhang, Q. Q., He, K., Zhang, Q., and Chai, L.: Sulfate-nitrate-ammonium aerosols over China: response to 2000–2015 emission changes of sulfur dioxide, nitrogen oxides, and ammonia, Atmos. Chem. Phys., 13, 2635-2652, 10.5194/acp-13-2635-2013, 2013.